# VETA-DiT: Variance-Equalized and Temporally Adaptive Quantization for Efficient 4-bit Diffusion Transformers

**Qinkai Xu**\* **Yijin Liu**\* **Yang Chen** **Lin Yang** **Li Li**† **Yuxiang Fu**†
Nanjing University
{qinkaixu, yijinliu, yangchen_nju}@smail.nju.edu.cn,
{linyang, lili, yuxiangfu}@nju.edu.cn
\*Equal Contribution. †Corresponding Author.

## Abstract

Diffusion Transformers (DiTs) have recently demonstrated remarkable performance in visual generation tasks, surpassing traditional U-Net-based diffusion models by significantly improving image and video generation quality and scalability. However, the large model size and iterative denoising process introduce substantial computational and memory overhead, limiting their deployment in real-world applications. Post-training quantization (PTQ) is a promising solution that compresses models and accelerates inference by converting weights and activations to low-bit representations. Despite its potential, PTQ faces significant challenges when applied to DiTs, often resulting in severe degradation of generative quality. To address these issues, we propose VETA-DiT (**V**ariance-**E**qualized and **T**emporal **A**daptation for **Di**ffusion **T**ransformers), a dedicated quantization framework for DiTs. Our method first analyzes the sources of quantization error from the perspective of inter-channel variance and introduces a Karhunen–Loève Transform enhanced alignment to equalize variance across channels, facilitating effective quantization under low bit-widths. Furthermore, to handle the temporal variation of activation distributions inherent in the iterative denoising steps of DiTs, we design an incoherence-aware adaptive method that identifies and properly calibrates timesteps with high quantization difficulty. We validate VETA-DiT on extensive image and video generation tasks, preserving acceptable visual quality under the more aggressive W4A4 configuration. Specifically, VETA-DiT reduces FID by 33.65 on the DiT-XL/2 model and by 45.76 on the PixArt-$\Sigma$ model compared to the baseline under W4A4, demonstrating its strong quantization capability and generative performance. Code is available at: https://github.com/xululi0223/VETA-DiT.

## 1 Introduction

Recently, Diffusion Transformers (DiTs) [33] have emerged as the dominant backbone architecture for diffusion models, replacing the U-Net structures [36]. They have been widely adopted in various generation tasks due to their superior performance [8, 30, 9]. A notable example is OpenAI's SoRA [32], which has attracted significant attention for its remarkable generation quality. Recent studies [11, 51] further demonstrate the impressive capability and scalability of DiTs across modalities.

Despite their success, DiTs face two major limitations: the inherently lengthy iterative denoising process and the growing model size, both of which lead to substantial computational and memory demands. These issues hinder the deployment of DiTs in resource-constrained environments and limit their applicability in real-time scenarios. For instance, generating a 1024×1024 resolution image with DiTs can take up to 10 seconds even on a NVIDIA A100 GPU.

39th Conference on Neural Information Processing Systems (NeurIPS 2025).

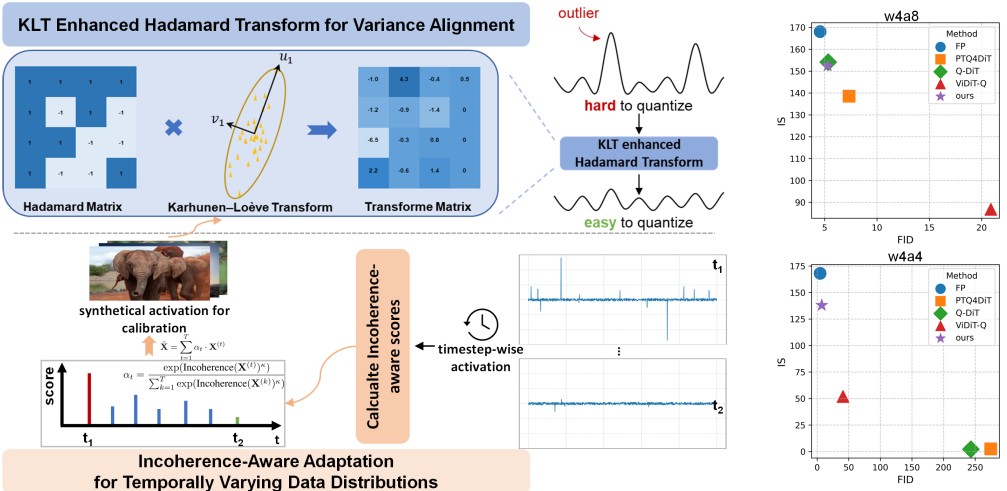

Figure 1: **(Left)** Overview of VETA-DiT. The Hadamard matrix is enhanced by a K-L transform to adapt to data distributions and reduce quantization difficulty. Calibration data fed into the KLT is synthesized via an incoherence-aware adaptive strategy to account for temporal variance across different timesteps. **(Right)** Quantization performance under W4A8 and W4A4, where points closer to the top-left corner indicate better performance. Please refer to Section 4.2 for detailed results.

Model quantization [31, 19] has been widely explored as an effective technique to accelerate inference by reducing memory footprint and computational overhead. It achieves this by compressing model weights and activations from floating-point to low-bit integer representations. However, the complex data distributions of weights and activations in DiTs pose significant challenges for existing quantization techniques, especially when targeting ultra-low bit-width without severely degrading performance. We identify two key challenges that hinder low-bit quantization of DiTs: (1) the presence of extreme outliers in specific channels results in significant inter-channel variance. Traditional approaches such as applying a smoothing factor [46, 45] or using simple Hadamard transforms [2, 49] fail to effectively align this variance, making low-bit quantization inaccurate; (2) the inherent iterative denoising nature of DiTs causes large variations in activation distributions across different timesteps, making it difficult to strike a balance between quantization effectiveness and computational efficiency.

To address these challenges, we propose a novel quantization method tailored for DiTs, termed VETA-DiT. We first analyze the limitations of directly applying the Hadamard transform from the perspective of inter-channel variance, and introduce a KLT enhanced Hadamard transform to achieve better variance alignment, enabling acceptable performance even under low-bit scenarios. To handle the temporal variation in activation distributions, we design an incoherence-aware adaptive strategy to identify the challenging timesteps for quantization. We then construct a synthetical calibration set that ensures both quantization accuracy and efficiency.

Our contributions are summarized as follows:

1. We analyze the limitations of directly applying the Hadamard transform in DiTs and propose a K-L transform-enhanced Hadamard method to effectively align inter-channel variance.

2. We investigate the quantization difficulty caused by temporal activation variation, and propose an incoherence-aware importance scoring strategy to construct a synthetical calibration set that captures representative distributions across different timesteps.

3. We conduct extensive experiments on both image and video generation tasks with multiple DiT models, and push the quantization precision to W4A4, demonstrating effectiveness of VETA-DiT in achieving high quantization performance without sacrificing visual fidelity.

## 2 Background and Related Works

### 2.1 Diffusion Transformer

Diffusion Models (DMs) [16] have attracted significant attention due to their powerful generative capabilities in visual domains such as image [33, 4] and video [29] synthesis. DMs simulate a

forward process that progressively adds Gaussian noise to the original data via a Markov chain [35], perturbing the data into a distribution close to standard Gaussian. A learned denoising network is then employed in the reverse process to iteratively reconstruct high-quality samples. The denoising process is typically parameterized by a deep neural network $\epsilon_\theta$, which predicts the noise at each timestep. Given a sample drawn from standard Gaussian noise $x_T \sim \mathcal{N}(0, I)$, the model iteratively denoises it through:

$$p_\theta(x_{t-1}|x_t) = N(x_{t-1}; \mu_\theta(x_t, t), \hat{\beta}_t I),$$ (1)

$$\mu_\theta(x_t, t) = \frac{1}{\sqrt{\alpha_t}}(x_t - \frac{1 - \alpha_t}{\sqrt{1 - \bar{\alpha}_t}}\epsilon_{\theta,t}), \quad \hat{\beta}_t = \frac{1 - \bar{\alpha}_{t-1}}{1 - \bar{\alpha}_t}.$$ (2)

The architecture of the noise prediction network plays a crucial role in determining the performance of diffusion models. DiT is a representative method that integrates Transformer architectures into diffusion models. Its core building block consists of multiple Transformer layers, each composed of a Multi-Head Self-Attention (MHSA) mechanism and a Feed-Forward Network (FFN) [42, 7, 33]. Specifically, MHSA and FFN are formulated as follows, respectively:

$$MHSA(\mathbf{Q}, \mathbf{K}, \mathbf{V}) = Softmax(\frac{\mathbf{Q}\mathbf{K}^T}{\sqrt{d_k}})\mathbf{V},$$ (3)

$$FFN(\mathbf{X}) = LayerNorm(\mathbf{X} + MLP(\mathbf{X})).$$ (4)

To incorporate conditional information (e.g., class labels), each Transformer block employs MLPs to project a condition vector $\mathbf{c} \in \mathbb{R}^{d_{in}}$ into scale and shift parameters, which are then injected into the hidden state $\mathbf{Z} \in \mathbb{R}^{n \times d_{in}}$ via an adaptive LayerNorm (adaLN) [33]:

$$(\gamma, \beta) = MLPs(\mathbf{c}), \quad adaLN(\mathbf{Z}) = LN(\mathbf{Z}) \odot (1 + \gamma) + \beta.$$ (5)

Although DiT achieves superior generation quality and expressive power compared to traditional architectures, its inference phase incurs high computational and memory costs due to the deeply stacked Transformer layers and iterative denoising process.

## 2.2 Model Quantization

Model quantization [19], is a widely adopted technique for reducing model size and accelerating inference. Quantization methods are typically categorized into Quantization-Aware Training (QAT) [50, 43, 12] and Post-Training Quantization (PTQ) [10, 46, 49, 45, 5, 20]. QAT integrates the quantization process into the training pipeline, often preserving model performance but requiring extensive computational resources and retraining time. In contrast, PTQ statically analyzes pretrained models using a small calibration dataset to estimate quantization parameters, offering advantages in deployment speed and computational efficiency.

A typical example is symmetric linear quantization, which can be expressed as:

$$x_{int} = clamp(\lfloor \frac{x}{s} \rceil - z, p_{min}, p_{max}),$$ (6)

where $x$ and $x_{int}$ denote the original and quantized data, respectively; $s$ is the scaling factor, $z$ is the zero point, $\lfloor \cdot \rceil$ denotes the rounding operation, and the clamp function constrains the quantized values within $[p_{min}, p_{max}]$.

Although PTQ techniques have demonstrated promising results on architectures such as CNNs [21, 22, 44] and ViTs [23, 44, 27], directly applying them to DiTs remains challenging. On one hand, DiTs often exhibit significant inter-channel distribution imbalance, making it difficult for a unified quantization range to accommodate all channels effectively. On the other hand, the stepwise sampling mechanism in diffusion processes introduces strong temporal dynamics in the activation distributions, necessitating quantization strategies that are adaptive to the timestep. To address these challenges, several studies have explored efficient PTQ methods for diffusion models. For instance, Q-Diffusion [21] and PTQ4DM [39] analyze activation variance across time steps to improve temporal robustness, while Q-DiT and PTQ4DiT incorporate joint temporal-channel characteristics into the design of adaptive quantization mechanisms. ViDiT-Q [49] proposes a unified approach that accounts for both temporal dependency and inter-channel imbalance by leveraging Hadamard rotations and asymmetric formats to enhance quantization effectiveness. However, they suffer from challenges under lower bit-width quantization settings. Recently, SVDQuant [20] employs a high-precision low-rank branch to take in the weight outliers with singular value decomposition, which is orthogonal to our method.

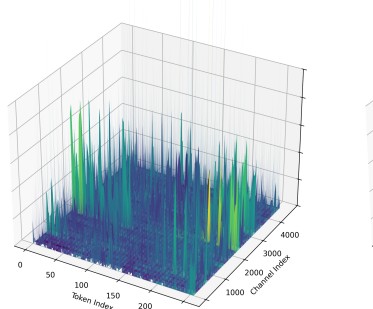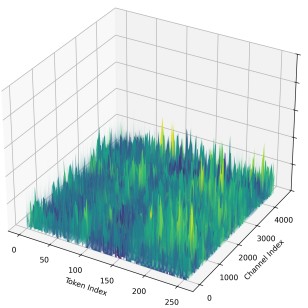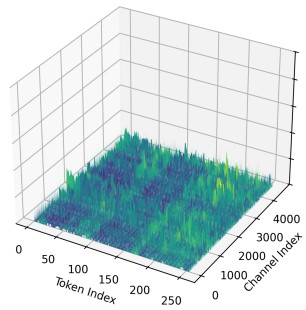

Figure 2: **(Left)** Activation distribution in the DiT model, showing extreme outliers in certain channels. Distributions after the Hadamard **(Middle)** and K-L-enhanced Hadamard transforms **(Right)**, respectively. While the Hadamard transform helps reduce variance imbalance, it has limitations; the K-L-enhanced method further improves inter-channel variance alignment.

## 3 Method

### 3.1 KLT Enhanced Hadamard Transform for Variance Alignment

We begin by analyzing the intermediate activations of various layers in the DiT-XL/2 model, focusing on selected linear layers to investigate their activation distribution characteristics. As shown in Figure 2 (Left), the activations exhibit a distribution pattern similar to those observed in LLMs and DMs [13, 39]: certain channels contain outlier values with extremely large magnitudes. In some layers, the magnitude of the outliers can be up to 100 times larger than the rest of the activations. These outliers significantly increase the variance within individual channels, resulting in substantial variance discrepancies across different channels [1, 2, 24]. Since per-token quantization methods assume uniform variance across channels, such discrepancies introduce considerable quantization errors, which are particularly problematic in hardware deployment.

Previous works in LLM quantization have proposed using the Hadamard transform to mitigate the influence of outliers and reduce inter-channel variance. Formally, given an input tensor $\mathbf{X} \in \mathbb{R}^{n \times m}$, where $n$ is the number of samples and $m$ is the number of channels, the Hadamard transform is defined as: $\mathbf{Z} = \mathbf{X}\mathbf{H}$, where $\mathbf{H} \in \mathbb{R}^{m \times m}$ is the Hadamard matrix. The Hadamard matrix $\mathbf{H}$ is an orthogonal matrix whose entries are drawn from $\left\{ +\frac{1}{\sqrt{m}}, -\frac{1}{\sqrt{m}} \right\}$. Let $\mathbf{z}_j$ denote the $j$-th column of $\mathbf{Z}$, i.e., the $j$-th transformed channel. Its variance is:

$$Var(\mathbf{z}_j) = \frac{1}{n} \sum_{i=1}^{n} z_{ij}^2 = \frac{1}{n} \sum_{i=1}^{n} \left( \sum_{k=1}^{m} x_{ik} h_{kj} \right)^2. \tag{7}$$

Utilizing the linearity of expectation and the properties of variance and covariance, we derive:

$$Var(\mathbf{z}_j) = \sum_{k=1}^{m} Var(\mathbf{x}_k) h_{kj}^2 + \sum_{k \neq l} Cov(\mathbf{x}_k, \mathbf{x}_l) h_{kj} h_{lj}. \tag{8}$$

This expression shows that the variance of the transformed channel consists of two components: a weighted sum of the original channel variances and a cross-covariance term. Since the cross-covariance term and the corresponding Hadamard weight vector $h_{:,j}$ vary across channels $j$, the resulting variances $Var(\mathbf{z}_j)$ cannot be assumed to be numerically similar in most cases. As a result, the transformed channels have unequal variances, limiting the effectiveness of the Hadamard transform for variance equalization in quantization. As shown in Figure 2 (Middle), the activations after the Hadamard transform alone still exhibit nonnegligible inter-channel variance, which adversely affects the quantization performance.

To achieve better variance alignment, we seek a transformation that ensures uniform average energy across all transformed channels. Let $v_k = Var(\mathbf{x}_k)$ be the variance of the $k$-th channel, and define the energy vector $\mathbf{v} = [v_1, v_2, \ldots, v_m]^T$. Our goal is to find a linear transformation $\mathbf{T} \in \mathbb{R}^{m \times m}$ such that the transformed tensor $\mathbf{Y} = \mathbf{X}\mathbf{T}$ satisfies:

$$Var(\mathbf{y}_j) = \frac{1}{m} \sum_{k=1}^{m} v_k, \quad \forall j. \tag{9}$$

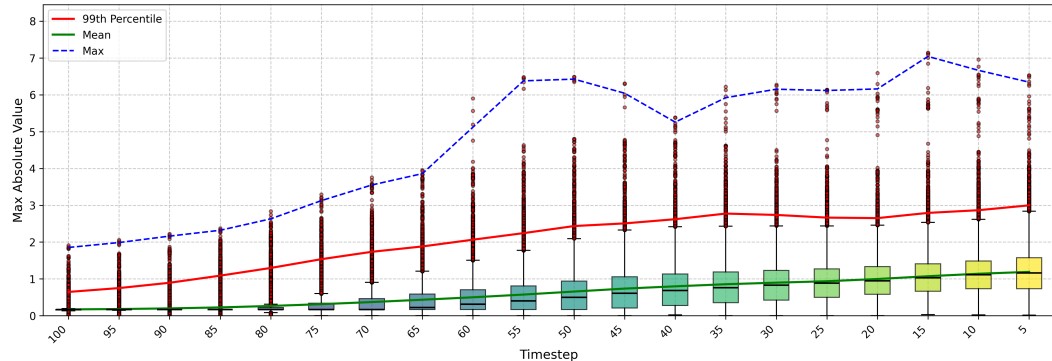

Figure 3: Boxplot of the maximum absolute activation values across channels at different timesteps in DiT, illustrating significant temporal variance that affects the effectiveness and efficiency of quantization methods.

That is, all output channels have the same average variance, maximizing the quantization range and minimizing quantization error.

This can be achieved using Karhunen–Loève Transform (KLT), which constructs an orthogonal basis from the eigenvectors of the input covariance matrix [6]. The KLT distributes total variance across orthogonal directions, minimizing redundancy and maximizing statistical independence. It has also been applied to improve quantization performance in state space models [48]. By combining KLT with the structured and computationally efficient Hadamard transform, we choose a composite transformation called the KLT enhanced Hadamard transform (KLT-H): $\mathbf{T}_{\text{KLT-H}} = \mathbf{KH}$, where $\mathbf{K}$ is the KLT matrix composed of eigenvectors of the input covariance matrix. Since both $\mathbf{K}$ and $\mathbf{H}$ are orthogonal, the composite transformation $\mathbf{T}_{\text{KLT-H}}$ remains orthogonal. Its inverse is simply $\mathbf{T}_{\text{KLT-H}}^{\top}$, which enables computational consistency by ensuring that the quantized model produces mathematically equivalent outputs to the original model, as illustrated by $(\mathbf{XT}_{\text{KLT-H}})(\mathbf{T}_{\text{KLT-H}}^{\top}\mathbf{W}) = \mathbf{XW}$. After transformation, the variance of each channel becomes:

$$Var(\mathbf{y}_j) = \sum_{k=1}^{m} \lambda_k h_{kj}^2, \tag{10}$$

where $\lambda_k$ are the eigenvalues of the input covariance matrix. For normalized Hadamard matrices, $h_{kj}^2 = \frac{1}{m}$, leading to:

$$Var(\mathbf{y}_j) = \frac{1}{m} \sum_{k=1}^{m} \lambda_k, \tag{11}$$

which indicates that the variance becomes equalized across different channels. After the KLT enhanced Hadamard transform, the inter-channel variance is further reduced, as illustrated in Figure 2 (Right). Thus, the transformed tensor achieves variance alignment across channels, significantly reducing quantization error and enhancing the robustness of diffusion model inference under low-bit quantization.

### 3.2 Incoherence-Aware Adaptation for Temporally Varying Data Distributions

DiT models generate images through an iterative denoising process. As shown in Figure 3, we observe that the activation distributions at different timesteps during the denoising process vary significantly. Previous quantization methods [26, 49] typically apply the same transformation matrix $\mathbf{H}$ to all timesteps within a layer. While this approach ensures computational efficiency, it often leads to severe performance degradation under low-bit quantization. Unfortunately, the K-L transformation matrix is highly sensitive to the distribution characteristics of the calibration dataset, and temporal variance in activation distributions undermine its effectiveness.

A straightforward solution is to compute a dedicated K-L transformation matrix for each timestep. However, this would incur substantial storage and computation overhead, counteracting the efficiency benefits of quantization.

To address this challenge, we propose an incoherence-aware, time-adaptive strategy that improves the performance of quantized models while preserving computational efficiency. The concept of

incoherence, adapted from [3, 41], is used to measure the degree of anomaly in activation data. Specifically, a matrix $\mathbf{X} \in \mathbb{R}^{m \times n}$ is said to be $\mu$-incoherent if for any $i$ and $j$:

$$|\mathbf{X}_{ij}| = |e_i^T \mathbf{X} e_j| \le \mu \cdot \frac{||\mathbf{X}||_F}{\sqrt{mn}}, \tag{12}$$

where $|| \cdot ||_F$ denotes the Frobenius norm. A higher incoherence value indicates that certain channel activations deviate significantly from the global mean, suggesting skewed distributions and outliers that complicate the quantization process.

During quantization, we assign each timestep $t$ an importance score $s_t$, which guides the KLT to pay more attention to activation data with highly skewed distributions. The score is defined as:

$$s_t = \text{Incoherence}(\mathbf{X}^{(t)}) = \frac{\max |\mathbf{X}^{(t)}|}{||\mathbf{X}^{(t)}||_F / \sqrt{mn}}, \tag{13}$$

where $\mathbf{X}^{(t)}$ denote the activation data sampled at timestep $t$. However, since incoherence values can differ drastically across timesteps, directly using $s_t$ may cause certain timesteps to dominate the calibration statistics. To mitigate this, we normalize the scores such that the sum of weights across all timesteps equals 1, ensuring a balanced contribution. The normalized form is given by:

$$\alpha_t = \frac{s_t^\kappa}{\sum_{k=1}^T s_k^\kappa}, \tag{14}$$

where $\kappa > 0$ is a control parameter. When $\kappa = 1$, the weighting is linear; higher values of $\kappa$ emphasize timesteps with extreme incoherence more strongly.

In practice, we apply an entropy-based regularization and a softmax-based weighting formulation, which together enhance stability and differentiability:

$$\alpha_t = \frac{\exp(\text{Incoherence}(\mathbf{X}^{(t)})^\kappa)}{\sum_{k=1}^T \exp(\text{Incoherence}(\mathbf{X}^{(k)})^\kappa)}. \tag{15}$$

We construct a synthetical calibration dataset $\tilde{\mathbf{X}}$ by combining activations across all timesteps with their respective weights:

$$\tilde{\mathbf{X}} = \sum_{t=1}^T \alpha_t \cdot \mathbf{X}^{(t)}. \tag{16}$$

In this way, the synthetical calibration dataset reflects activation distribution characteristics across multiple timesteps. This enables more accurate variance alignment, reduces quantization error, and improves the robustness of the quantized model. We apply the proposed method to the linear layer analyzed in Figure 3, where the incoherence-based importance scores across timesteps are computed. As shown in Figure 4 (Left), these scores exhibit a strong positive correlation with the inter-channel variance at each timestep, indicating that the method effectively prioritizes timesteps with higher quantization difficulty. We then perform KLT enhanced Hadamard transforms using calibration datasets constructed from either randomly sampled timestep activations or the incoherence-aware aggregation. The resulting channel-wise variance distributions, shown in Figure 4 (Right), demonstrate that the incoherence-aware approach effectively adapts to temporal variance and significantly reduces inter-channel variance.

## 4 Experiments

### 4.1 Experimental Settings

**Image Generation.** We first evaluate our proposed VETA-DiT framework on the image generation task. Following the original evaluation settings of DiT [33], we use the pretrained DiT-XL/2 model [33] to generate images with resolutions of $256 \times 256$ on the ImageNet dataset [37]. The DDIM-solver [40] is employed during generation, with sampling steps set to 50 and 100. To further demonstrate the generality of VETA-DiT on diverse generation tasks, we also integrate it into the PixArt-$\Sigma$ model [4] for prompt-based image generation on the COCO dataset [25]. A DPM-solver [28] with 20 steps is used, the classifier-free guidance (CFG) scale is set to 4.5. We use FID [15], spatial FID(sFID),

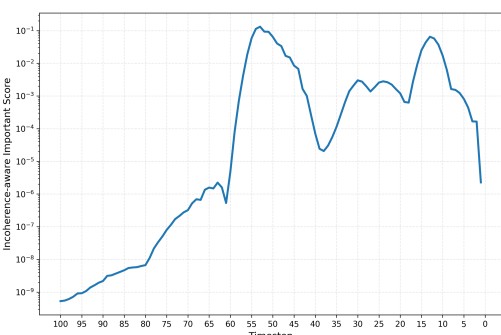 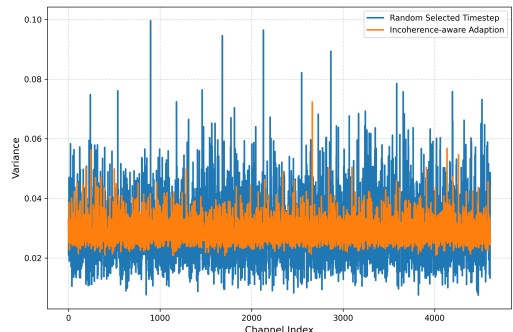

Figure 4: **(Left)** Incoherence-aware importance scores across timesteps, showing a positive correlation with inter-channel variance at each timestep. **(Right)** Channel-wise variance distributions obtained by applying the K-L enhanced Hadamard transform using calibration datasets constructed from either randomly sampled timestep activations or incoherence-weighted aggregated activations. The incoherence-aware adaptation effectively accommodates temporal variation.

Inception Score [38], and Precision to evaluate generation fidelity. Additionally, for the COCO task, we include ClipScore [14] to assess text-image alignment, and ImageReward [47] to estimate human preference under complex prompt conditions.

**Video Generation.** We further evaluate the effectiveness of VETA-DiT in the video generation task by integrating it into the STDiT3 model from the Open-Sora [17]. Videos are generated using a 50-step DDIM-solver with a CFG scale of 4.0. We conduct comprehensive evaluations on the VBench benchmark to obtain detailed quantitative results. Following [49, 34], we select 8 major evaluation dimensions from VBench [18].

## 4.2 Quantization Performance

We conduct a comprehensive evaluation of our proposed VETA-DiT and compare it against several state-of-the-art post-training quantization (PTQ) approaches specifically designed for DiTs under various experimental settings. In particular, we consider three representative baselines: PTQ4DiT [45], Q-DiT [5], and ViDiT-Q [49]. To ensure fair and consistent comparisons, we re-implement and adapt these methods based on their official open-source repositories to support different network architectures and task scenarios. For ViDiT-Q, we intentionally exclude the use of its original mixed-precision quantization strategy in our experiments, as it tends to quantize the majority of linear layers to INT8, which could obscure the performance differences among these methods. Further experimental details can be found in Appendix Section A.

**Image Generation Results.** Table 1 presents the quantitative results of DiT-XL/2 on the ImageNet 256×256 resolution task. Specifically, PTQ4DiT and Q-DiT achieve acceptable performance under the W4A8 setting, benefiting from their inter-channel importance balancing and dynamic group-wise quantization strategies. Our proposed VETA-DiT method further narrows the performance gap with the full-precision (FP) model under the same setting, achieving a best FID of 5.22 and an Inception Score of 152.52, indicating higher-quality image generation. When the quantization setting is further reduced to W4A4, PTQ4DiT and Q-DiT fail to produce meaningful images due to the limitations of their methodologies. Remarkably, VETA-DiT remains effective in generating visually acceptable images even under this low-precision setting, reducing FID and sFID by 33.65 and 16.68, respectively, compared to the second-best method. This demonstrates the effectiveness of our proposed variance-balancing strategy and temporal difference-aware adaptation. ViDiT-Q, without the use of mixed-precision quantization, which is incompatible with many existing hardware platforms, suffers from severe performance degradation. We further evaluate VETA-DiT on the COCO dataset using the PixArt-$\Sigma$ model for text-to-image generation. As shown in Figure 5, the results are consistent with those observed on ImageNet: ViDiT-Q fails to generate acceptable outputs under W4A8 due to the lack of mixed-precision support. Although Q-DiT performs reasonably well at W4A8, it struggles with handling inter-channel imbalances, leading to poor-quality generated images. In contrast, VETA-DiT consistently delivers superior performance across both bit-width settings. Additional randomly generated images are provided in Appendix Figure 10, 11, 12, 13 for visual comparison.

Table 1: Performance comparison of DiT-XL/2 on the ImageNet 256×256.

| Model | Bit-width | Method | IS(↑) | FID(↓) | sFID(↓) | Precision(↑) |
|-------|-----------|--------|-------|--------|---------|--------------|
| DiT-XL/2 steps=100 cfg=1.0 | W16A16 | FP | 90.32 | 12.25 | 19.28 | 0.65 |
| | W4A8 | PTQ4DiT | 66.85 | 17.34 | 20.81 | 0.58 |
| | | Q-DiT | **77.07** | 16.16 | 19.24 | 0.61 |
| | | ViDiT-Q | 37.46 | 49.65 | 31.87 | 0.41 |
| | | **Ours** | 75.90 | **15.22** | **18.66** | **0.62** |
| | W4A4 | PTQ4DiT | 2.08 | 304.62 | 129.68 | 0.07 |
| | | Q-DiT | 1.93 | 256.13 | 429.47 | 0.01 |
| | | ViDiT-Q | 25.11 | 72.23 | 44.57 | 0.32 |
| | | **Ours** | **64.91** | **22.85** | **20.50** | **0.57** |
| DiT-XL/2 steps=100 cfg=1.5 | W16A16 | FP | 168.08 | 4.56 | 17.72 | 0.79 |
| | W4A8 | PTQ4DiT | 138.54 | 7.33 | 22.40 | 0.74 |
| | | Q-DiT | **154.13** | 5.34 | 17.62 | 0.76 |
| | | ViDiT-Q | 86.81 | 20.87 | 25.40 | 0.56 |
| | | **Ours** | 152.53 | **5.29** | **17.50** | **0.77** |
| | W4A4 | PTQ4DiT | 2.42 | 274.59 | 117.25 | 0.08 |
| | | Q-DiT | 2.14 | 243.05 | 410.74 | 0.02 |
| | | ViDiT-Q | 51.96 | 40.87 | 35.58 | 0.45 |
| | | **Ours** | **137.98** | **7.22** | **18.90** | **0.74** |
| DiT-XL/2 steps=50 cfg=1.0 | W16A16 | FP | 88.07 | 13.38 | 19.07 | 0.65 |
| | W4A8 | PTQ4DiT | 61.08 | 23.41 | 22.29 | 0.54 |
| | | Q-DiT | **76.12** | 17.42 | 18.92 | 0.61 |
| | | ViDiT-Q | 30.79 | 58.48 | 39.57 | 0.36 |
| | | **Ours** | 75.05 | **17.30** | **18.36** | **0.63** |
| | W4A4 | PTQ4DiT | 2.05 | 298.57 | 126.64 | 0.08 |
| | | Q-DiT | 1.81 | 262.58 | 421.08 | 0.01 |
| | | ViDiT-Q | 18.13 | 84.40 | 57.46 | 0.28 |
| | | **Ours** | **64.15** | **24.24** | **21.98** | **0.56** |
| DiT-XL/2 steps=50 cfg=1.5 | W16A16 | FP | 164.61 | 4.86 | 17.65 | 0.80 |
| | W4A8 | PTQ4DiT | 129.06 | 8.89 | 22.73 | 0.71 |
| | | Q-DiT | **148.83** | 5.77 | **17.57** | 0.76 |
| | | ViDiT-Q | 81.65 | 22.65 | 26.45 | 0.56 |
| | | **Ours** | 147.12 | **5.53** | 17.65 | **0.77** |
| | W4A4 | PTQ4DiT | 2.33 | 281.43 | 119.45 | 0.08 |
| | | Q-DiT | 2.02 | 249.98 | 404.09 | 0.01 |
| | | ViDiT-Q | 48.34 | 44.15 | 37.90 | 0.43 |
| | | **Ours** | **135.66** | **7.90** | **19.11** | **0.73** |

'(WxYb)' indicates that the weights and activations are quantized to x-bit and y-bit, respectively.

| Bit-width | Method | IS(↑) | FID(↓) | sFID(↓) | CLIP(↑) | IR(↑) |
|-----------|--------|-------|--------|---------|---------|-------|
| W16A16 | FP | 38.27 | 59.74 | 294.91 | 0.27 | 0.89 |
| W4A8 | Q-DiT | **38.34** | 63.36 | 299.88 | 0.26 | **0.93** |
| | ViDiT-Q | 18.81 | 125.32 | 325.12 | 0.22 | -0.55 |
| | **Ours** | 37.62 | **60.46** | **297.20** | 0.26 | 0.90 |
| W4A4 | Q-DiT | 18.27 | 107.98 | 312.25 | 0.24 | -0.43 |
| | ViDiT-Q | 9.70 | 227.80 | 332.97 | 0.20 | -1.52 |
| | **Ours** | **35.56** | **65.22** | **305.36** | **0.26** | **0.88** |

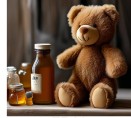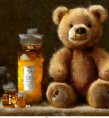

Figure 5: **(Left)** Text-to-image generation performance of the PixArt-Σ model on the COCO dataset. **(Right)** Generated images comparison of W4A4 quantization.

**Video Generation Results..** As shown in Table 2, similar to the image generation task, existing quantization methods achieve satisfactory performance under the W4A8 setting. However, due to their limited ability to balance inter-channel variance, they struggle under the more challenging W4A4 setting. In contrast, our method consistently outperforms these baselines across various metrics, demonstrating its effectiveness in preserving both the visual quality and temporal consistency of generated videos. Appendix Figure 14, 15 presents a visual comparison of a randomly generated video for visual evaluation. We also conduct additional experiments to further evaluate the video generation performance of the quantized models; see Appendix D.3 for details.

Table 2: Performance comparison of Open-Sora on the VBench evaluation benchmark.

| Bit-width | Method | Subject Consist. | BG. Consist. | Motion Smooth. | Dynamic Degree | Aesthetic Quality | Imaging Quality | Scene Consist. | Overall Consist. |
|---|---|---|---|---|---|---|---|---|---|
| W16A16 | FP | 0.947 | 0.965 | 0.983 | 0.680 | 0.575 | 0.519 | 0.454 | 0.274 |
| W4A8 | Q-DiT | **0.951** | **0.962** | 0.985 | 0.600 | 0.569 | 0.497 | 0.451 | 0.273 |
|  | ViDiT-Q | 0.920 | 0.963 | 0.982 | 0.520 | 0.536 | 0.495 | 0.316 | 0.269 |
|  | Ours | 0.950 | 0.961 | **0.985** | **0.600** | **0.571** | **0.509** | **0.506** | **0.273** |
| W4A4 | Q-DiT | 0.934 | 0.955 | 0.983 | 0.440 | 0.508 | 0.452 | 0.311 | 0.252 |
|  | ViDiT-Q | 0.897 | 0.962 | 0.978 | 0.600 | 0.499 | 0.468 | 0.333 | 0.254 |
|  | Ours | **0.939** | **0.959** | **0.984** | **0.480** | **0.546** | **0.499** | **0.431** | **0.269** |

## 4.3 Ablation Studies

To assess the effectiveness of each component in our proposed framework, we conduct ablation experiments under the challenging W4A4 setting. These experiments are carried out using the DiT-XL/2 model on the ImageNet 256×256 dataset, with 50 sampling steps using the DDIM-solver. Detailed quantitative results are provided in Table 3. We begin our evaluation with a baseline that employs a simple group-wise linear quantization strategy. Under the W4A4 configuration, the baseline performs poorly across all metrics. Next, we incorporate a K-L enhanced Hadamard transform for variance alignment, which significantly improves the quality of generated images, achieving a FID of 48.32, and an IS of 43.68. Building upon this, we further introduce the incoherence-aware temporal adaptation method, which enhances the ability of the Hadamard transform to adapt across different timesteps. This results in a further reduction of the FID to 7.90. These findings demonstrate the individual effectiveness of each proposed component and highlight how their integration contributes to pushing our VETA-DiT method toward state-of-the-art performance under W4A4 settings, enabling the generation of visually plausible and semantically coherent images.

Table 3: Ablation study on ImageNet 256×256 with W4A4.

| Method | IS | FID | sFID | Precision |
|---|---|---|---|---|
| FP | 164.61 | 4.86 | 17.64 | 0.80 |
| Baseline | 1.84 | 260.47 | 409.15 | 0.01 |
| + K-L enhanced Hadamard Transform | 43.68 | 48.32 | 40.19 | 0.36 |
| + Incoherence-aware adaption | 135.66 | 7.90 | 19.10 | 0.72 |

## 5 Conclusion

This paper presents **VETA-DiT**, a post-training quantization framework designed for Diffusion Transformers (DiTs) to reduce inference cost while preserving generation quality. We address two major challenges: large inter-channel variance due to outliers and significant activation distribution shifts across denoising timesteps. To this end, we introduce a KLT-enhanced Hadamard transform for variance alignment and an incoherence-aware adaptive calibration method to handle temporal dynamics. Experiments on image and video generation tasks show that VETA-DiT achieves performance close to the full-precision model under W4A8, while still delivering acceptable visual quality under the W4A4 setting. Our approach advances the deployment of DiTs in resource-constrained scenarios.

# 6 Acknowledgement

This work was supported in part by the National Key Research and Development Program of China under Grant No. 2023YFB2806802 and Grant No. 2021YFB3600104, in part by the Joint Funds of the National Nature Science Foundation of China under Grant No. U21B2032 and in part by Suzhou's "Jiebang Guashuai" Project for Key Core Technologies under Grant No. SYG2024134. Lin Yang would like to thank the support from NSFC (No. 62306138), JiangsuNSF (No. BK20230784), and the Inovation Program of State Key Laboratory for Novel Software Technology at Nanjing University (No. ZZKT2024B15, ZZKT2025B25). The authors are grateful for the help from the Interdisciplinary Research Center for Future Intelligent Chips (Chip-X) and Yachen Foundation.

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

# A  Additional Experimental Details

In our implementation, we randomly selected 16 classes from the ImageNet dataset to perform image generation and calibrate the DiT-XL/2 model. For the PixArt-$\Sigma$ model, we used the sample prompts provided in the official PixArt-$\Sigma$ codebase for calibration. In video generation tasks, we calibrated the model using text-to-video prompts from the Open-Sora implementation.

For weight quantization, we adopted static quantization with a group size of 128. Additionally, we applied second-order information from the Hessian matrix, as used in GPTQ, to adjust the weights. This process is performed offline and introduces no additional overhead during inference. For activation quantization, we followed the same dynamic quantization strategy as Q-DiT, using a group size of 128 to balance quantization accuracy and inference cost. All experiments were conducted on Nvidia A800 GPUs.

For all baseline methods, we used the same calibration set, random noise, and classifier-free guidance, and quantized all Transformer layers in the DiT architecture.

In the ViDiT-Q implementation, we collected the maximum activation values of the layers to be quantized in order to determine the smooth factor. We set the hyperparameter $\alpha = 0.5$ to control the smooth factor. The original ViDiT-Q applies mixed-precision quantization to both weights and activations, which we found leads to use higher precision in the linear layers of the FFN module. To ensure a fair comparison, we removed mixed-precision support in our implementation of ViDiT-Q and used the same precision settings as all other methods.

# B  Detailed Algorithm for VETA-DiT

---

**Algorithm 1:** KLT-H Based Quantization with Incoherence-Aware Temporal Sampling

---

**Input:** Activation tensor $\mathbf{X} \in \mathbb{R}^{t \times n \times m}$, weight matrix $\mathbf{W} \in \mathbb{R}^{m \times d}$
**Output:** Quantized activation tensor $\mathbf{X}_q$, quantized weight matrix $\mathbf{W}_q$

1   **Step 1: Incoherence-aware sampling of activations**
2   Initialize an importance vector $\boldsymbol{\alpha}$ of length $t$;
3   **for** *each time step $t_i$* **do**
4       Extract the activation matrix $\mathbf{X}^{(t_i)}$ at time $t_i$;
5       Compute the incoherence score $s_{t_i}$ of $\mathbf{X}^{(t_i)}$;
6       Compute important score $\alpha_{t_i}$ based on $s_{t_i}$;
7       Store $\alpha_{t_i}$ in $\boldsymbol{\alpha}$;

8   **Step 2: Generate sampling distribution**
9   Apply softmax to $\boldsymbol{\alpha}$ to obtain probability vector $\boldsymbol{p}$;
10   Sample multiple time steps based on $\boldsymbol{p}$;
11   Concatenate the sampled activation matrices to form $\mathbf{X}_{\text{sampled}}$;

12   **Step 3: Construct KLT-H transformation matrix**
13   Compute the covariance matrix of $\mathbf{X}_{\text{sampled}}$ and derive its eigenvectors to form the KLT matrix $\mathbf{K}$;
14   Construct the Hadamard matrix $\mathbf{H}$;
15   Form the composite orthogonal transform: $\mathbf{T}_{\text{KLT-H}} = \mathbf{K} \cdot \mathbf{H}$;

16   **Step 4: Transform activation and weight using shared transform**
17   Transform activation: $\mathbf{X}' = \mathbf{X} \cdot \mathbf{T}_{\text{KLT-H}}$;
18   Transform weight: $\mathbf{W}' = \mathbf{T}_{\text{KLT-H}}^{\top} \cdot \mathbf{W}$;
19   This ensures that the computation $(\mathbf{X} \cdot \mathbf{W})$ is equivalent to $(\mathbf{X}' \cdot \mathbf{W}')$, preserving functional correctness;

20   **Step 5: Quantize the transformed activation and weight**
21   Quantize $\mathbf{X}'$ using an dynamic activation quantization method, producing $\mathbf{X}_q$;
22   Quantize $\mathbf{W}'$ using a static weight quantization method, producing $\mathbf{W}_q$;

23   **return** *Quantized activation $\mathbf{X}_q$ and quantized weight $\mathbf{W}_q$*

---

The VETA-DiT pipeline is detailed in Algorithm 1. The algorithm consists of five key steps. **Step 1**, we compute the importance scores for each diffusion time step using incoherence-aware sampling. Time steps with higher quantization difficulty are assigned larger weights according to Equation (13, 14), allowing the transformation matrix to adapt to the temporally varying activation distributions. **Step 2**, after obtaining the importance scores, we enhance the numerical stability of the sampling process by applying Equation (15, 16) to generate a composite calibration set, which includes activations sampled in proportion to their importance. **Step 3**, we calculate the eigenvectors and eigenvalues of the covariance matrix of the incoherence-aware calibration set to construct the Karhunen–Loève Transform (KLT) matrix, which is then combined with a randomly generated Hadamard matrix, to form the KLT-enhanced Hadamard transformation matrix. **Step 4**, we apply the transformation matrix $\mathbf{T}_{\text{KLT-H}}$ to the activation tensor and its inverse (the transpose of the matrix due to orthogonality) to the weight tensor, aligning the inter-channel variance of both activations and weights while ensuring the correctness of the computation. **Step 5**, we apply dynamic quantization to the transformed activations and static quantization to the transformed weights, resulting in low-bit quantized activations and weights for efficient inference.

## C  Proofs

This section provides a detailed derivation demonstrating that the Hadamard transform enhanced by the K-L transform can achieve variance equalization across channels. Recall that the variance of the $j$-th transformed channel can be expressed as Equation 7. Expanding the squared summation, we obtain:

$$Var(\mathbf{z}_j) = \frac{1}{n} \sum_{i=1}^{n} \sum_{k=1}^{m} \sum_{l=1}^{m} x_{ik} x_{il} h_{kj} h_{lj}. \tag{17}$$

Interchanging the summation order and grouping terms:

$$Var(\mathbf{z}_j) = \sum_{k=1}^{m} \sum_{l=1}^{m} h_{kj} h_{lj} \left( \frac{1}{n} \sum_{i=1}^{n} x_{ik} x_{il} \right). \tag{18}$$

We define the sample covariance matrix $\mathbf{C} \in \mathbb{R}^{m \times m}$ as:

$$C_{kl} = \frac{1}{n} \sum_{i=1}^{n} x_{ik} x_{il}. \tag{19}$$

Then the variance of the $j$-th transformed channel becomes:

$$Var(\mathbf{z}_j) = \sum_{k=1}^{m} \sum_{l=1}^{m} C_{kl} h_{kj} h_{lj} = \mathbf{h}_j^\top \mathbf{C} \mathbf{h}_j, \tag{20}$$

where $\mathbf{h}_j \in \mathbb{R}^m$ is the $j$-th column of the orthogonal matrix $\mathbf{H}$.

To further interpret this expression, recall that:

- The diagonal elements of $\mathbf{C}$ are $C_{kk} = Var(\mathbf{x}_k)$.
- The off-diagonal elements are $C_{kl} = Cov(\mathbf{x}_k, \mathbf{x}_l)$ for $k \neq l$.

Thus, we can split the summation in Equation 20 into diagonal and off-diagonal parts to derive Equation 8, which shows even after applying an orthogonal transformation like the Hadamard transform, the resulting channel variances may still differ substantially due to the interaction between original variances and covariances. Hence, Hadamard transform alone does not guarantee uniform variance across channels.

Karhunen–Loève Transform (KLT), which uses the eigenvectors of the covariance matrix $\mathbf{C}$ to form an orthogonal basis. Let $\mathbf{C} = \mathbf{U}\mathbf{\Lambda}\mathbf{U}^\top$, where $\mathbf{U}$ contains the orthonormal eigenvectors and $\mathbf{\Lambda}$ is a diagonal matrix of eigenvalues $\lambda_1, \lambda_2, ..., \lambda_m$. Define $\mathbf{K} = \mathbf{U}^\top$ as the KLT matrix. The KLT transform is then $\mathbf{X}\mathbf{K}$, with covariance:

$$\mathbf{C}_{\text{KLT}} = \frac{1}{n} (\mathbf{X}\mathbf{K})^\top (\mathbf{X}\mathbf{K}) = \mathbf{K}^\top \mathbf{C} \mathbf{K} = \mathbf{\Lambda}. \tag{21}$$

Thus, the transformed features are decorrelated and their variances are exactly the eigenvalues $\lambda_1, \ldots, \lambda_m$.

To combine the statistical decorrelation of KLT with the computational efficiency of Hadamard, we obtain the transformed tensor $\mathbf{Y} = \mathbf{XKH} = \mathbf{XT}_{\text{KLT-H}}$.

This combines the fast computation of the Hadamard transform with the statistical decorrelation property of the KLT. The covariance matrix of the transformed output $\mathbf{Y}$ is:

$$\mathbf{C_Y} = \frac{1}{n}\mathbf{Y}^\top\mathbf{Y} = \mathbf{H}^\top\mathbf{K}^\top\mathbf{CKH} = \mathbf{H}^\top\mathbf{\Lambda H}. \tag{22}$$

Then, the variance of the $j$-th channel in $\mathbf{Y}$ is:

$$Var(\mathbf{y}_j) = \mathbf{h}_j^\top\mathbf{\Lambda}\mathbf{h}_j = \sum_{k=1}^{m}\lambda_k h_{kj}^2. \tag{23}$$

For a normalized Hadamard matrix, $h_{kj}^2 = \frac{1}{m}$ for all $k, j$, so we can obtain the Equation 11.

## D  Additional Empirical Results

### D.1  Validation of KLT-H Transformation Effectiveness

We further analyze the effectiveness of the proposed KLT-H transformation in terms of incoherence reduction for both activations and weights. Specifically, we compute the normalized incoherence values across all layers of the DiT-XL/2 Transformer backbone under three settings: without transformation (original), with Hadamard transformation, and with our proposed KLT-H transformation. As shown in Figure 6 and Figure 7, the KLT-H consistently achieves lower incoherence than the other two settings, indicating its superior ability to decorrelate channels and better align with the intrinsic data structure. These improvements hold consistently for both weights (Figure 6) and activations (Figure 7), further validating the generality of our approach across different components of the model.

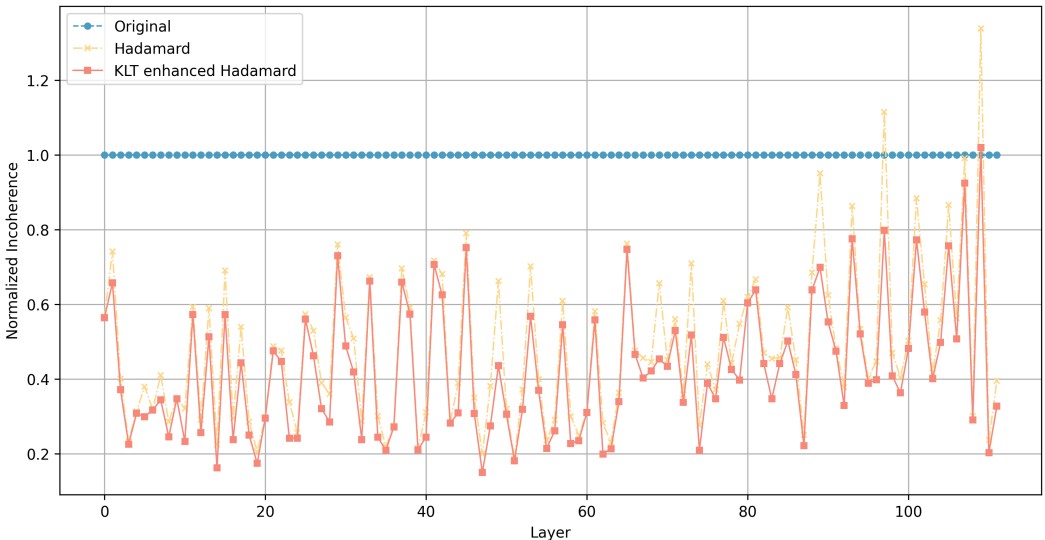

Figure 6: Normalized incoherence values of weights across all layers in the DiT-XL/2 Transformer backbone under different transformations. The KLT-H transformation consistently achieves lower incoherence, indicating its effectiveness in improving weight channel alignment.

In addition to the layer-wise results, we also compute the average incoherence and standard deviation across all layers. As summarized in Table 4, the KLT-H transformation yields the lowest average incoherence and the smallest standard deviation, further demonstrating its effectiveness and robustness

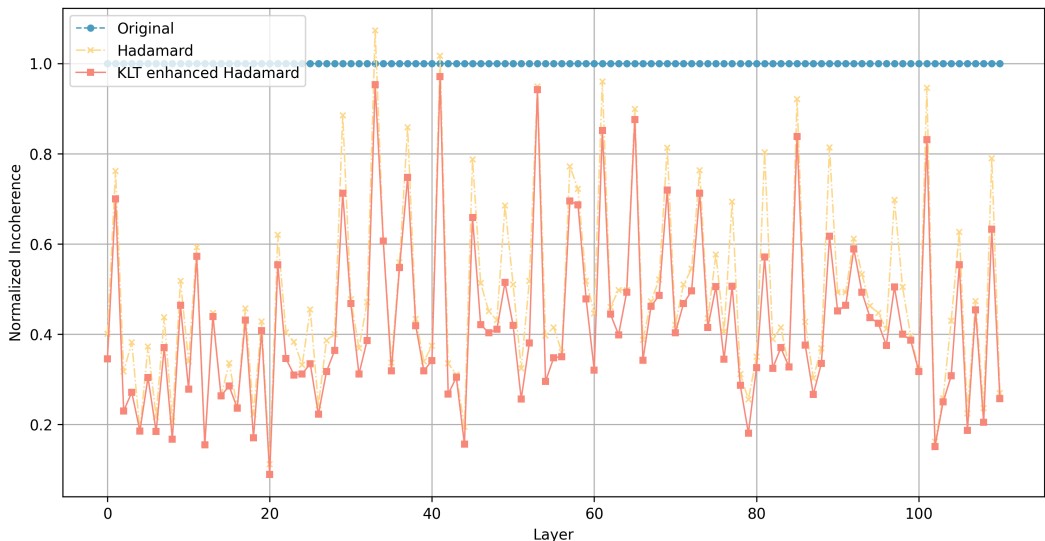

Figure 7: Normalized incoherence values of activations across all layers in the DiT-XL/2 Transformer backbone under different transformations. The KLT-H transformation significantly reduces incoherence compared to both the original and Hadamard-transformed activations, demonstrating improved inter-channel decorrelation.

in suppressing inter-channel redundancy. These results provide strong empirical evidence supporting the use of KLT-H as a principled and practical enhancement for quantization under temporally varying data distributions.

Table 4: Average and standard deviation of incoherence values across all layers of the DiT-XL/2 Transformer under different transformations.

| Method | Weight-Incoherence | Activation-Incoherence |
| --- | --- | --- |
| Original | $19.91 \pm 13.26$ | $49.29 \pm 44.05$ |
| Hadamard | $8.56 \pm 5.56$ | $19.22 \pm 11.80$ |
| KLT enhanced Hadamard | $7.42 \pm 4.40$ | $17.13 \pm 10.74$ |

## D.2 Evaluation of Incoherence-Aware Adaptation Strategy

To further validate the effectiveness of our proposed incoherence-aware adaptation strategy, we conduct additional experiments comparing three methods: (1) the standard RTN baseline without any transformation, (2) the KLT-H transform constructed using randomly selected samples, and (3) our incoherence-aware KLT-H transform. For each method, we compute the reconstruction error (mean squared error, MSE) between the output of each linear layer and the corresponding full-precision output.

As shown in Figure 8, under the W4A8 setting, the incoherence-aware KLT-H transform consistently achieves lower reconstruction errors across the majority of layers compared to the other two baselines. Similarly, Figure 9 presents the results under the W4A4 setting, where the advantage of our method becomes even more pronounced due to the more aggressive quantization constraints. These results highlight the robustness of the incoherence-aware adaptation in handling temporally varying data distributions under low-bit quantization.

## D.3 Evaluation of Temporal and Textual Consistency in Video Generation

To comprehensively evaluate the generative capability of the quantized models, we additionally adopt three metrics from different perspectives. Specifically, CLIPSIM and CLIP-Temp are used to measure

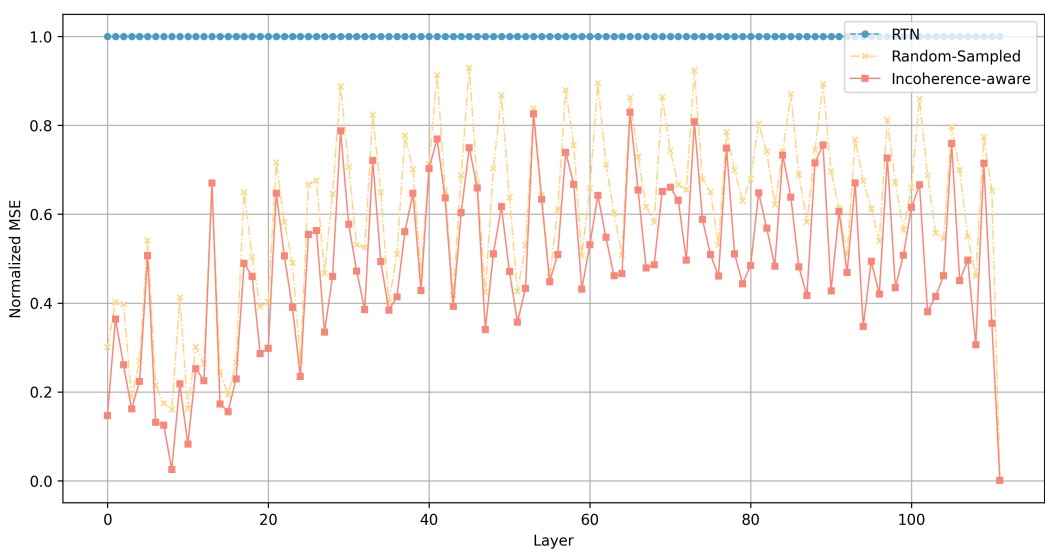

Figure 8: Normalized layer-wise MSE reconstruction error comparison under W4A8 quantization.

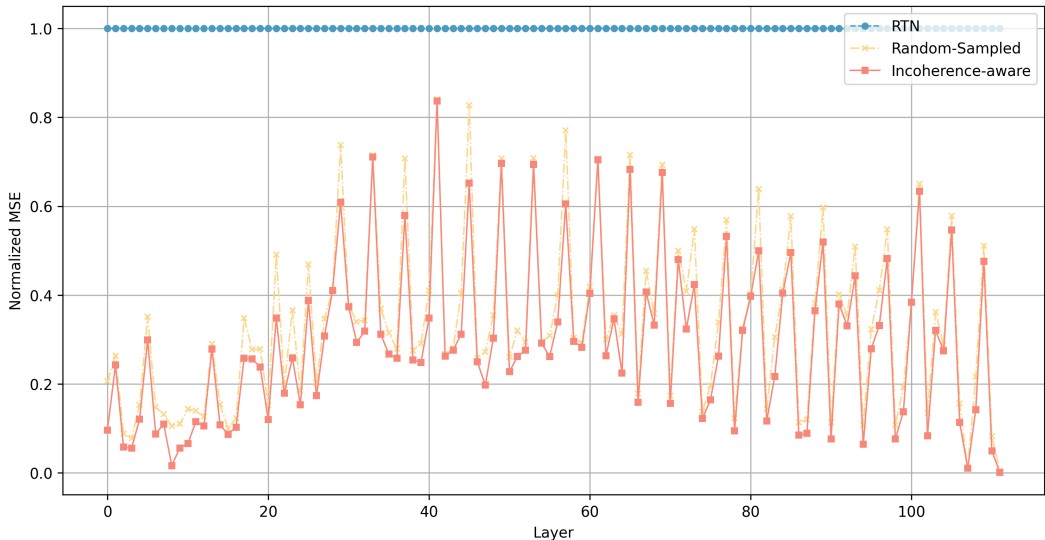

Figure 9: Normalized layer-wise MSE reconstruction error comparison under W4A4 quantization.

text-video alignment and temporal semantic consistency, respectively, while Temporal Flickering is employed to assess temporal smoothness. As shown in Table 5, our method demonstrates consistently strong performance under both W4A8 and W4A4 quantization settings, outperforming other PTQ methods in terms of both textual alignment and temporal consistency.

# E    Limitations and Broader Impacts

In this work, we present an effective post-training quantization framework that promotes the broader deployment and applicability of Diffusion Transformers (DiTs). By leveraging variance equalization and low-bit data representation, our method significantly reduces the computational and memory overhead, thereby improving the usability of DiTs in resource-constrained settings. VETA-DiT relies on a carefully designed incoherence-aware sampling strategy that assigns higher importance to timesteps with higher quantization difficulty. However, its applicability to other domains or modalities—such as audio and 3D data—requires further investigation. In future work, we plan

Table 5: Comparison of video generation quality under W4A8 and W4A4 settings using CLIPSIM, CLIP-Temp, and Temporal Flickering.

| Bit-Width | Method | CLIPSIM | CLIP-Temp | Temporal Flick |
|---|---|---|---|---|
| W16A16 | FP | 0.2217 | 0.9980 | 0.9782 |
| W4A8 | Q-DiT | 0.2198 | 0.9973 | 0.9798 |
| | ViDiT-Q | 0.2200 | 0.9975 | **0.9820** |
| | **Ours** | **0.2204** | **0.9982** | 0.9804 |
| W4A4 | Q-DiT | 0.2161 | 0.9952 | 0.9702 |
| | ViDiT-Q | 0.2156 | 0.9953 | 0.9762 |
| | **Ours** | **0.2176** | **0.9974** | **0.9770** |

to explore hardware acceleration of VETA-DiT to enable real-time inference. In terms of broader impacts, this work advances efficient generative modeling by improving quantization techniques for diffusion models. Nonetheless, as with many optimization techniques that facilitate model deployment in low-resource scenarios, there is a risk of misuse in surveillance, deepfakes, or other sensitive content generation without proper regulation. We encourage the community to adopt responsible practices that align with ethical AI development principles.

# F    Additional Visualization Results

In this section, we present additional qualitative results to further demonstrate the effectiveness of our proposed VETA-DiT. Figure 10, 11 shows randomly generated ImageNet images using the DiT-XL/2 model under W4A8 and W4A4 quantization settings. Figure 12, 13 also provides examples of images generated by the PixArt-Sigma model. In addition, a frame sample from a randomly generated video is shown in Figure 14, 15. These results qualitatively highlight the strong generation capability of VETA-DiT under both moderate and aggressive quantization.

# G    Inference overhead

The KLT-enhanced Hadamard matrix is defined as $\mathbf{T}_{\text{KLT-H}} = \mathbf{KH}$, where $\mathbf{K}$ is a data-driven KLT matrix, precomputed offline using the calibration set, and $\mathbf{H}$ is a random Hadamard matrix.

During inference, this transformation can be fused into model weights, incurring no additional cost for most layers. For layers like out-proj and down-proj, online transformation is necessary to preserve computational invariance. We evaluated the associated overhead on DiT-XL/2 via implementing a custom CUDA kernel on NVIDIA A100 GPU, targeting matrix multiplication operations within these layers.

Table 6: Comparison of latency, speedup, and memory reduction.

| Bit-Width | Method | Latency (ms) | Speedup | Memory (MB) / Reduction |
|---|---|---|---|---|
| W16A16 | FP | $6.167 \pm 0.045$ | - | 38.53 / - |
| W4A8 | ViDiT-Q | $3.879 \pm 0.012$ | $1.59\times$ | 18.68 / $2.06\times$ |
| W4A4 | Ours | $2.960 \pm 0.006$ | **2.08**$\times$ | 9.68 / **3.98**$\times$ |
| W4A4 + Online Transform | Ours | $3.163 \pm 0.007$ | $1.95\times$ | 10.31 / $3.74\times$ |

As shown in Table 6, due to the availability of efficient Hadamard transform implementations, the online transform introduces only 6.8% overhead, while still achieving $1.95\times$ acceleration and $3.74\times$ memory savings over FP16.

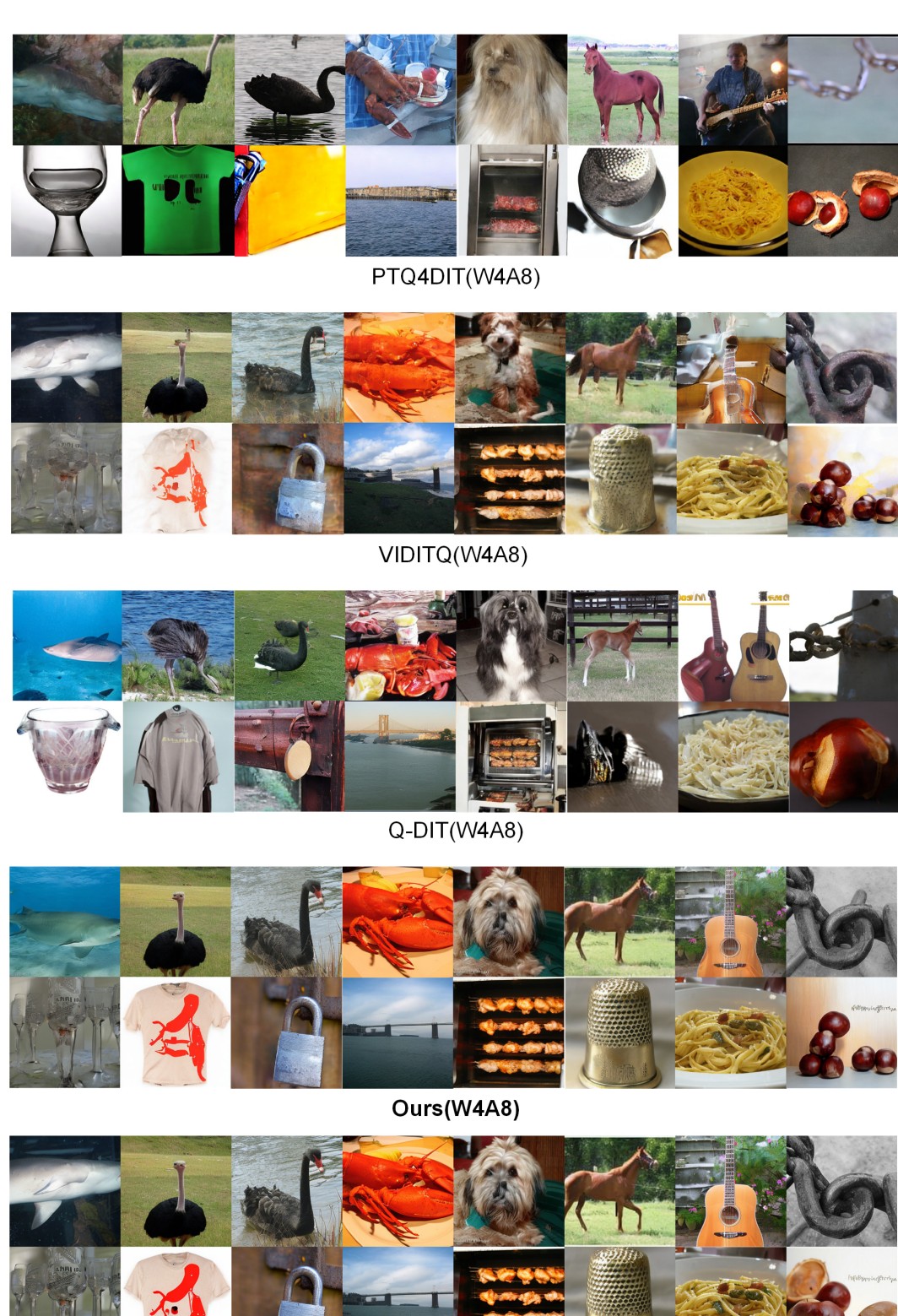

PTQ4DIT(W4A8)

VIDITQ(W4A8)

Q-DIT(W4A8)

**Ours(W4A8)**

Full-Precision

Figure 10: Random images generated under W4A8 quantization using different PTQ methods with the DiT-XL/2 model on ImageNet 256×256.

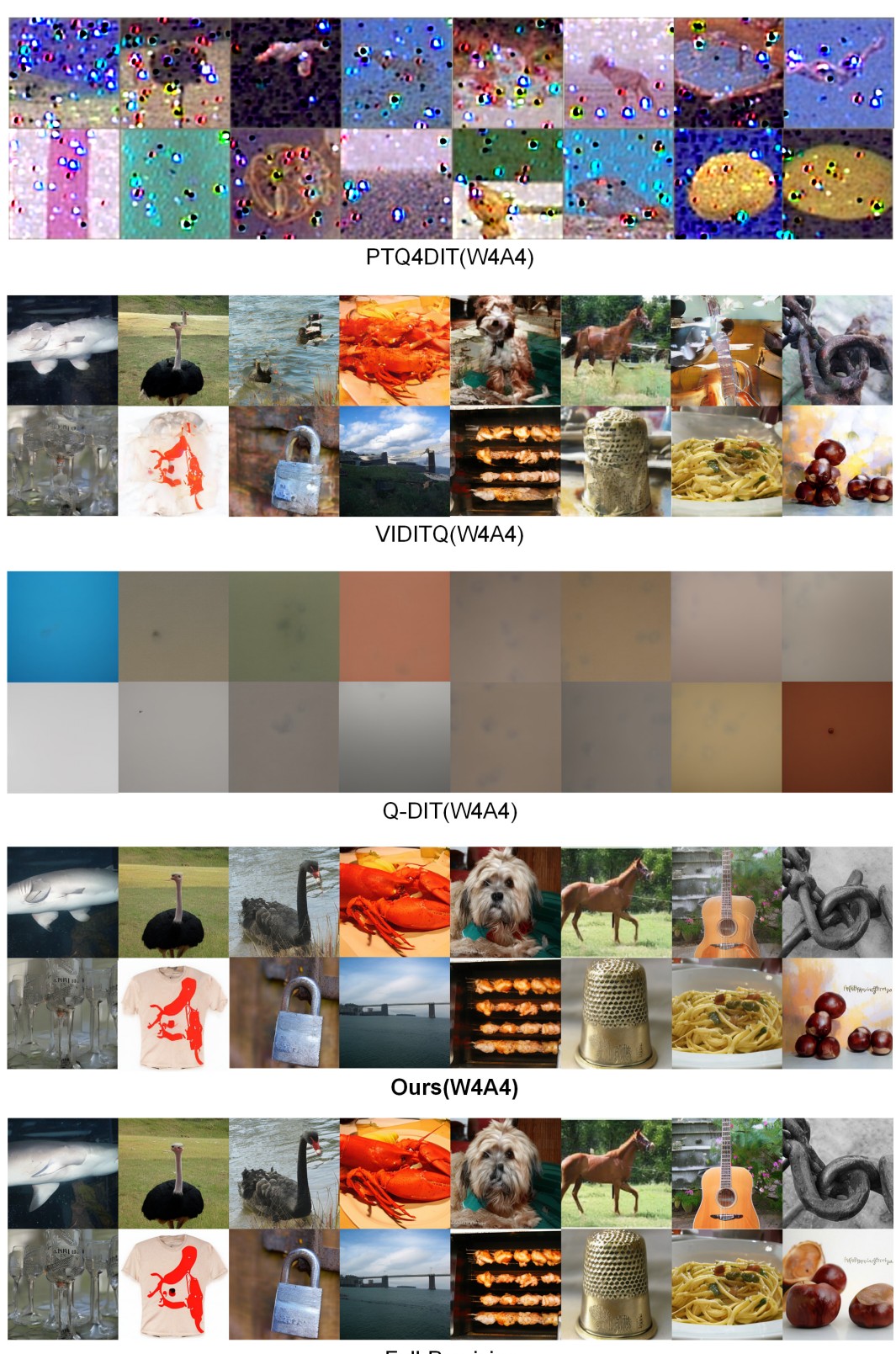

Figure 11: Random images generated under W4A4 quantization using different PTQ methods with the DiT-XL/2 model on ImageNet 256×256.

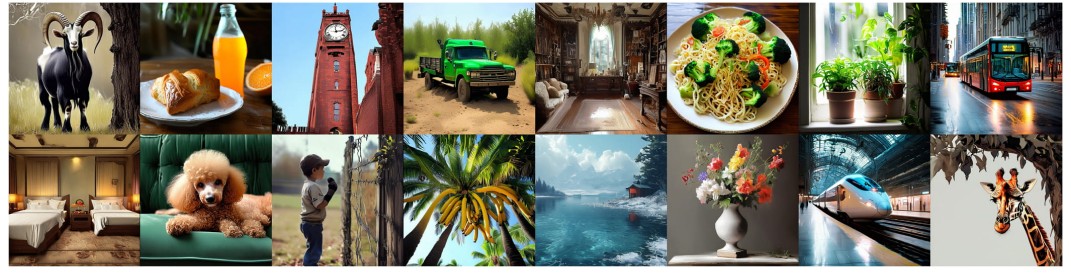

VIDITQ(mixed-presicion*)

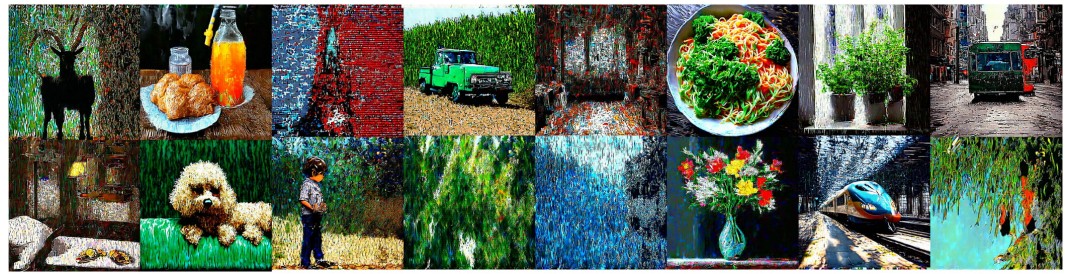

VIDITQ(W4A8)

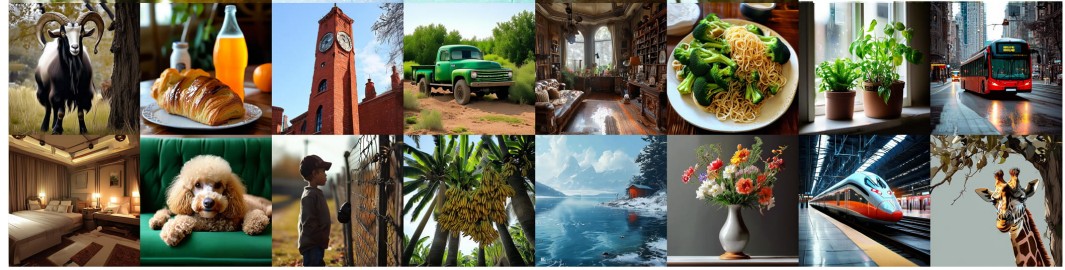

Q-DIT(W4A8)

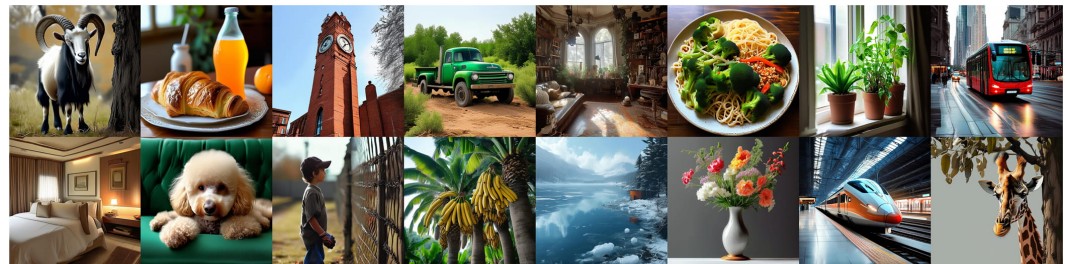

**Ours(W4A8)**

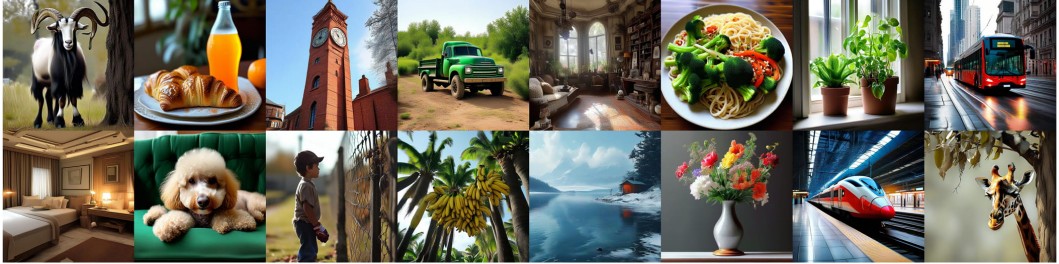

Full-Precision

Figure 12: Random images generated under W4A8 quantization using different PTQ methods with the Pixart-Σ model on COCO. * indicates that ViDiT-Q uses mixed-precision quantization, where most linear layers are quantized with higher bit-widths.

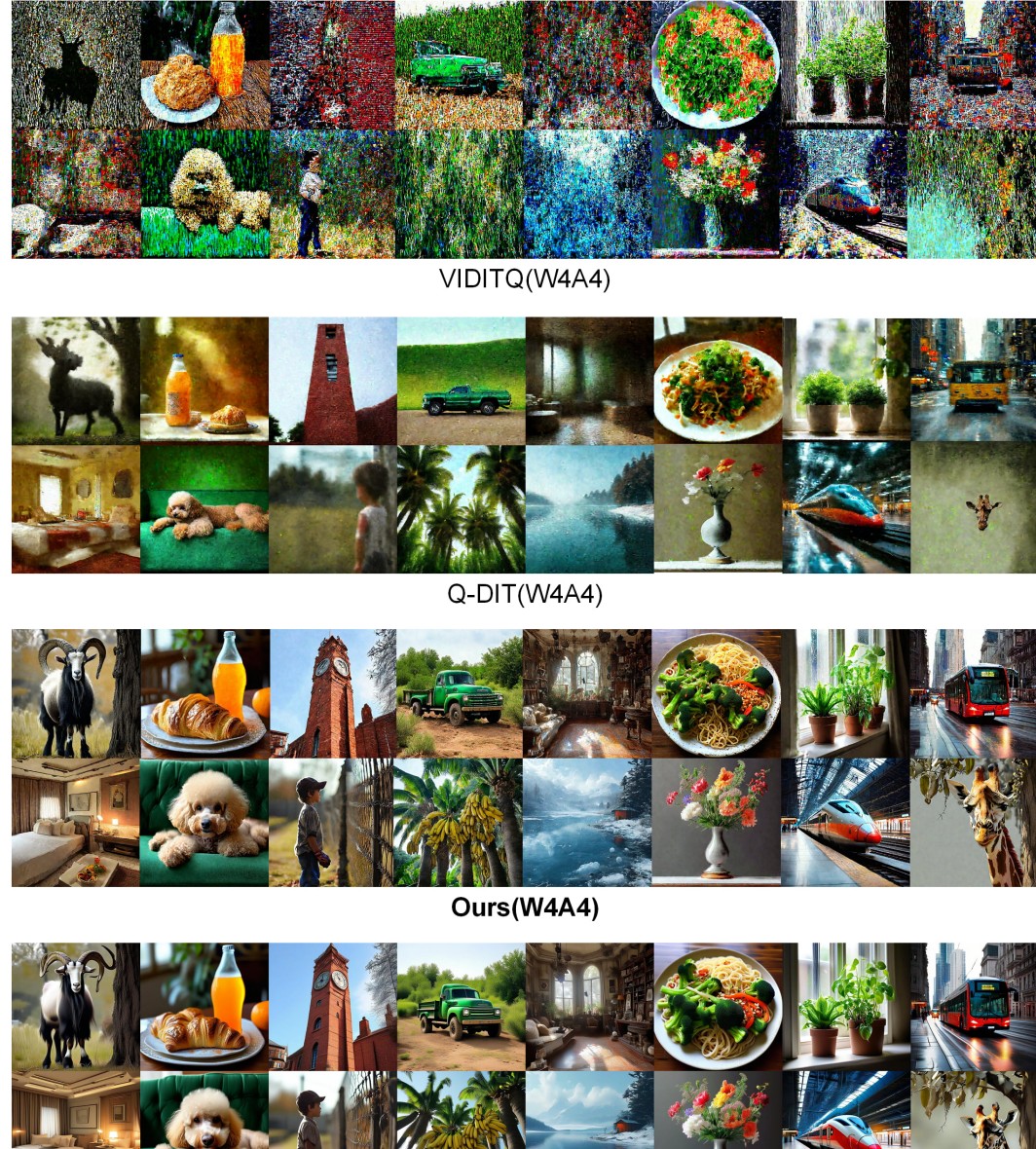

VIDITQ(W4A4)

Q-DIT(W4A4)

**Ours(W4A4)**

Full-Precision

Figure 13: Random images generated under W4A4 quantization using different PTQ methods with the Pixart-$\Sigma$ model on COCO.

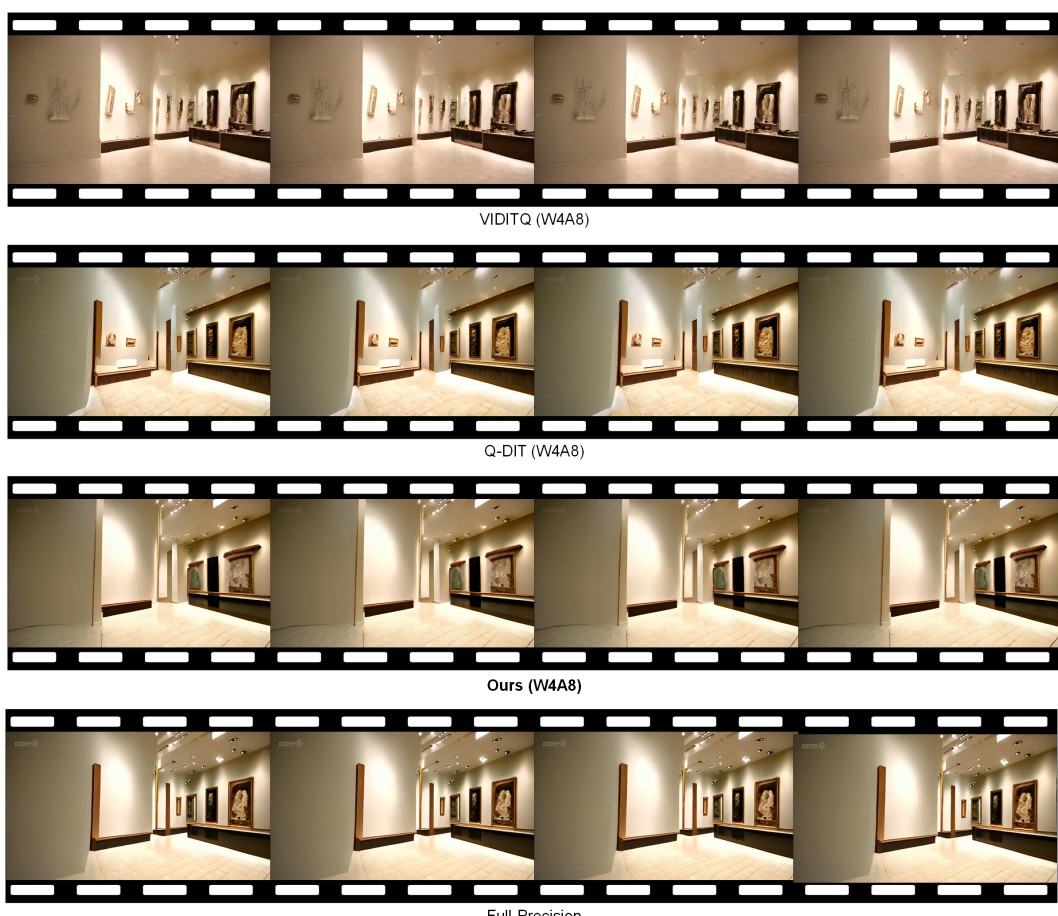

Figure 14: Random video generated under W4A8 quantization using different PTQ methods with the Open-Sora 1.2 model on VBench.

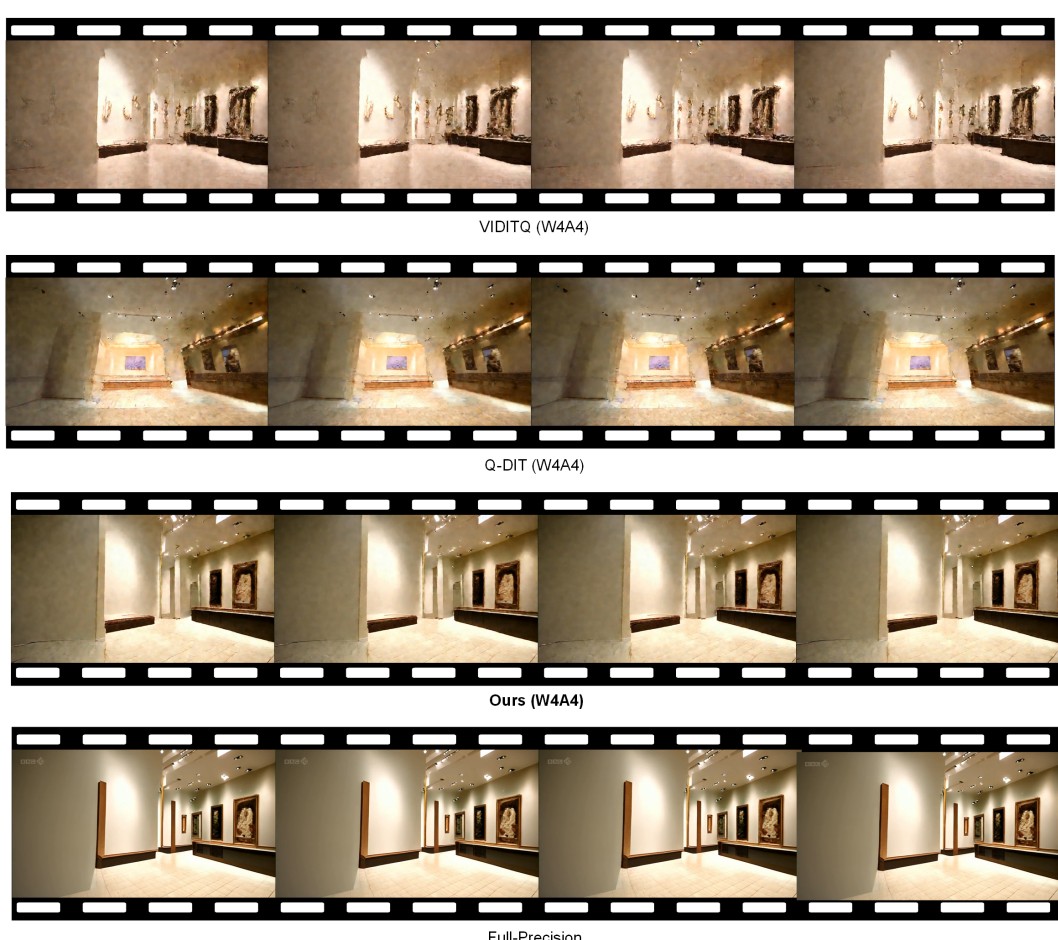

Figure 15: Random video generated under W4A4 quantization using different PTQ methods with the Open-Sora 1.2 model on VBench.

