# OpenReview forum: "VETA-DiT: Variance-Equalized and Temporally Adaptive Quantization for Efficient 4-bit Diffusion Transformers"
_NeurIPS.cc/2025/Conference — NeurIPS 2025 poster_

### Official Review · Reviewer_5kuo · 2025-06-24

**Clarity:** 3
**Significance:** 3
**Originality:** 3
**Rating:** 5
**Confidence:** 2

**Summary:**

This paper identifies two key challenges in quantizing DiT models: (1) large inter-channel variance caused by outliers, and (2) significant shifts in activation distributions across denoising timesteps. To address these issues, the authors propose a KLT-enhanced Hadamard transform for variance alignment and an incoherence-aware adaptive calibration method to manage temporal dynamics. Experimental results demonstrate strong performance compared to existing baselines.

**Questions:**

Q1: Can you provide more details on the practical implementation aspects of your method, especially regarding hardware compatibility or deployment on resource-constrained devices?

Q2: Given the clear performance differences observed when quantizing from W4A8 to W4A4, do you anticipate similar trends at lower bit-widths? What challenges do you foresee when pushing the quantization limits further?

**Ethical Concerns:**

["NO or VERY MINOR ethics concerns only"]

**Final Justification:**

The authors have addressed my main concerns. I think it is reasonable to keep the current score as accept.

**Limitations:**

yes

**Paper Formatting Concerns:**

There are no major formatting issues.

**Quality:**

3

**Strengths And Weaknesses:**

## Strengths
* **S1**: The paper is clearly written and easy to follow. The presentation is organized and the explanations are generally accessible, which helps the reader understand the main ideas without much difficulty.
* **S2**: The proposed method is simple and straightforward, making it easy to implement. This practical aspect increases its usability and potential for adoption in real-world applications.
* **S3**: The experimental results are strong and well-supported. The model is evaluated across multiple datasets, tasks, and metrics, showing its effectiveness and generality. Notably, the clearer performance gap observed when further quantizing from W4A8 to W4A4 provides additional evidence of the proposed method’s effectiveness.

## Weaknesses:
* **W1**: For a quantization method like this, I would expect an analysis of hardware resource usage. Unfortunately, the paper does not include such an evaluation.
* **W2**: The paper could be further strengthened by evaluating the proposed method at even lower bit settings, such as W2A8 or W4A2.

---

> ### Author Rebuttal · Authors · 2025-07-29
>
> Thank you sincerely for your thoughtful and positive feedback on our work. Below, we have provided a detailed explanation for your remaining concern as follows. Please do not hesitate to let us know if you have any further questions.
>
> **W1 & Q1**: Hardware resource usage evaluation.
>
> >We thank the reviewer for pointing this out. We have updated the manuscript to include an evaluation of hardware resource usage. In DiTs, linear layers dominate both computation and memory consumption [1, 2]. To address this, we implement our INT4 matrix multiplications using CUTLASS on Tensor Cores (CUDA 12.4) and evaluate our custom kernels on NVIDIA A100 GPU.
> >
> > We fuse most KLT matrices into model weights during offline calibration, thereby eliminating the need for runtime transformation. For layers that require runtime KLT to preserve mathematical equivalence [3], we use fast online Hadamard kernels, which are widely available and efficient.
> >
> >Despite incorporating both online quantization and runtime KLT transforms, we still observe **1.95$ \times$ speedup** and **3.74$\times$ memory savings** over FP16. Compared to the W4A8 implementation in ViDiT-Q, which adopts higher activation precision to preserve generation quality, our approach provides **1.23$\times$ speedup** and **1.81$\times$ memory reduction**, demonstrating the hardware efficiency of our method:
> >
> >|Bit-Width|Method|Latency (ms)|Speedup|Memory (MB)|Reduction|
> >|---|---|---|---|---|---|
> >|W16A16|FP|6.167 ± 0.045|-|38.53|-|
> >|W4A8|ViDiT-Q|3.879 ± 0.012| 1.59$\times$|18.68|2.06$\times$|
> >|W4A4|Ours|2.960 ± 0.006|**2.08$\times$**|9.68|**3.98$\times$**|
> >|W4A4 + Online Transform|Ours|3.163 ± 0.007|**1.95$\times$**|10.31|**3.74$\times$**|
> >
> >We believe this provides a comprehensive view of both computational and memory efficiency. We will add the results to the revised manuscript to provide a comprehensive view of runtime and memory efficiency.
> >
> >[1] Solving Oscillation Problem in Post-Training Quantization Through a Theoretical Perspective. CVPR 2023.
> >
> >[2] PTQ4DiT: Post-training Quantization for Diffusion Transformers. NeurIPS 2024.
> >
> >[3] SliceGPT: Compress Large Language Models by Deleting Rows and Columns. ICLR 2024.
>
> **W2 & Q2**: Evaluation under lower bit settings.
>
> >We conducted additional experiments on DiT-XL/2 under the W3A3 quantization setting, which is already highly challenging. As shown below, our method significantly outperforms Q-DiT, but performance degradation is still notable under such low bit-widths. Introducing mixed-precision, by keeping some sensitive layers in 8-bit, yields substantial quality improvements.
> >
> >| Bit-Width             | Method | IS $ \uparrow$    | FID $\downarrow$    | sFID $\downarrow$   | Precision  $\uparrow$|
> >| --------------------- | ------ | ------ | ------ | ------ | --------- |
> >| W16A16                | FP     | 164.61 | 4.86   | 17.65  | 0.80      |
> >| W3A3                  | Q-DiT  | 1.65   | 266.97 | 402.47 | 0.01      |
> >| W3A3                  | Ours   | 45.83  | 55.42  | 34.42  | 0.32      |
> >| W3A3(Mixed Precision)| **Ours**   | **111.62** | **13.71**  | **22.25**  | **0.66**      |
> >
> >At even lower bit-widths, quantizing becomes extremely lossy due to limited representational range. In such cases, offline KLT-based variance equalization becomes insufficient, and **online KLT computation may be required to maintain generative fidelity**. However, this would introduce non-negligible runtime cost. We will explore this trade-off in future work.

---

> > ### Comment · Reviewer_5kuo · 2025-08-03
> > **Response to Authors' rebuttal**
> >
> > Thank you for the detailed rebuttal. I appreciate the authors’ clarifications, which adequately address the main concerns. I continue to find the paper acceptable and maintain my score.

---

> > > ### Author Response · Authors · 2025-08-04
> > >
> > > Dear Reviewer,​​
> > >
> > > We sincerely appreciate your constructive comments and valuable suggestions.
> > >
> > > We are pleased to address all your questions and deeply grateful for your recognition of our work. Thank you again for your contributions to improving the quality of our manuscript.
> > >
> > > ​​Best, Authors

---

### Official Review · Reviewer_eHfZ · 2025-06-30

**Clarity:** 4
**Significance:** 2
**Originality:** 2
**Rating:** 4
**Confidence:** 4

**Summary:**

This paper proposes VETA-DiT, a post-training quantization framework for Diffusion Transformers (DiTs) under W4A4 setting. The method addresses two major challenges, large inter-channel variance due to outliers and timestep-dependent activation shifts during iterative denoising. To mitigate these, the authors introduce a KLT enhanced Hadamard transform for better variance alignment and an incoherence-aware temporal adaptation strategy. VETA-DiT outperforms existing PTQ baselines across image and video generation tasks, achieving high visual quality under low-bit quantization settings without retraining.

**Questions:**

1. The combination of KLT with the Hadamard transform appears similar to the method proposed in MambaQuant (ICLR 2025). Can the authors clearly articulate how their usage is different in either formulation, application, or motivation specific to DiTs?

2. While the paper addresses temporal variance in image generation (across denoising steps), it is unclear whether similar temporal adaptation is used or needed in video synthesis. Do video generation tasks show similar incoherence trends, and is the adaptation equally effective?

3. How does your incoherence-aware adaptation with synthetic data generation compare to previous methods built upon KLT-Hadamard transforms, in terms of effectiveness and novelty?

4. What is the actual runtime improvement or slowdown introduced by your method, including the overhead of the KLT-H computation and calibration?

5. You mention disabling mixed-precision in ViDiT-Q for fair comparison. Could you elaborate on the implications of this choice, and whether similar mixed-precision designs could benefit VETA-DiT as well?

**Ethical Concerns:**

["NO or VERY MINOR ethics concerns only"]

**Final Justification:**

During the rebuttal process, the authors provided additional supporting evidence. I think the novelty is somewhat limited, but the paper is above the decision boundary.

**Limitations:**

Yes

**Paper Formatting Concerns:**

No.

**Quality:**

3

**Strengths And Weaknesses:**

Strengths
1. The paper is well-structured, with clear visualizations and an accessible algorithm description, aiding reproducibility and understanding.
2. This paper integrates the Karhunen–Loève Transform (KLT) with the Hadamard transform in the context of DiT for the first time, and demonstrates outstanding generative quality in low-precision settings.
3. The incoherence-aware temporal adaptation and synthetic data generation strategy are straightforward and well-validated through diverse empirical experiments.
4. The proposed scheme is applied to large-scale networks for both image and video generation, demonstrating its generality and robustness.

Weaknesses
1. The combination of KLT with the Hadamard transform has already been presented in MambaQuant (ICLR 2025), which predates this NeurIPS submission. Thus, the novelty of this paper is somewhat reduced, although its contribution to the DiT domain remains valid and meaningful.
2. As a result, the main contribution of this work lies in the design of the incoherence-aware adaptation. While the idea is interesting, comparisons with alternative temporal calibration or sampling strategies are insufficient.
3. Since the method combines KLT with the Hadamard transform, the computational overhead is likely higher than using Hadamard alone. The paper should provide practical validation of inference speedups on real hardware.
4. While the paper analyzes the time-varying activation distributions for image generation, it gives limited attention to similar temporal characteristics in video synthesis tasks.

---

> ### Author Rebuttal · Authors · 2025-07-29
>
> We greatly appreciate the reviewer's constructive comments on our paper. We will respond to the reviewer's feedback with detailed explanations for each point.
>
> **W1 & Q1**: Difference from MambaQuant in the use of KLT.
>
> >We also noticed that MambaQuant applies KLT to state space models, as described in line 147 of our paper. MambaQuant focuses on Mamba-based state space models and addresses the numerical instability caused by outlier amplification in PScan operations. In contrast, our work is dedicated to Transformer-based diffusion models (DiTs), which exhibit a **naturally large variance across activation dimensions** and **significant temporal variation** due to iterative denoising.
> >
> >We introduce KLT to equalize the intrinsic variance distribution across channels and time steps in DiTs. However, applying a single, time-step-independent KLT matrix across all denoising steps, as MambaQuant, leads to a **severe degradation in generation quality** due to the significant temporal variation in DiT activations. On the other hand, using fully time-step-specific KLT matrices incurs excessive load/offload overhead during inference. To address this trade-off, we propose an incoherence-aware adaptation strategy that enables **efficient and temporally-sensitive KLT usage** in DiTs, effectively mitigating both representational and computational inefficiencies.
> >
> >We will include a more explicit discussion of the differences between our approach and MambaQuant in the revised manuscript.
>
> **W2**: Effectiveness compared to alternative temporal sampling strategies.
> >Prior methods, such as ViDiT-Q [1], adopt a shared balancing mask $s$ across all denoising steps, which fails to account for the dynamically changing data distributions inherent in DiTs, effectively corresponding to uniform timestep sampling without adaptation. Currently, there is a lack of viable strategies specifically designed to handle this temporal variation.
> >
> >Therefore, we provide a comprehensive evaluation of our incoherence-aware adaptation in Table 3 and Appendix D.2 of our paper, assessing both its impact on **final generation quality and its effectiveness in reducing quantization error during inference**. The results demonstrate that **our method significantly outperforms uniform and random sampling strategies**.
> >
> >[1] ViDiT-Q: Efficient and Accurate Quantization of Diffusion Transformers for Image and Video Generation. ICLR 2025.
>
> **W3 & Q4**: Runtime and calibration overhead.
>
> >The calibration process is conducted **once offline and does not affect inference latency**. On NVIDIA A100 GPUs, calibrating DiT-XL/2 takes approximately **35 minutes**, and calibrating Open-Sora v1.2 (1.1B parameters) takes around **2.5 hours**.
> >
> >During inference, we use CUTLASS INT4 kernels optimized for Tensor Cores (CUDA 12.4) and **fuse most KLT matrices into the model weights**. Only a few blocks (e.g., out-proj, down-proj) require lightweight online transforms, implemented via fast Hadamard kernels commonly available in existing libraries. We still achieve **1.95$ \times$ speedup** and **3.74$\times$ memory savings** over FP16. Compared to the W4A8 implementation in ViDiT-Q, which adopts higher activation precision to preserve generation quality, our approach provides **1.23$\times$ speedup** and **1.81$\times$ memory reduction**, demonstrating the hardware efficiency of our method:
> >
> >|Bit-Width|Method|Latency (ms)|Speedup|Memory (MB)|Reduction|
> >|---|---|---|---|---|---|
> >|W16A16|FP|6.167 ± 0.045|-|38.53|-|
> >|W4A8|ViDiT-Q|3.879 ± 0.012| 1.59$ \times$|18.68|2.06$\times$|
> >|W4A4|Ours|2.960 ± 0.006|**2.08$\times$**|9.68|**3.98$\times$**|
> >|W4A4 + Online Transform|Ours|3.163 ± 0.007|**1.95$\times$**|10.31|**3.74$\times$**|
> >
> >These results confirm that our method maintains efficiency while incorporating KLT-H adaptations. We will add the results to the revised manuscript to provide a comprehensive view of runtime and memory efficiency.
>
> **W4 & Q2**: Temporal incoherence in video generation.
>
> >Temporal incoherence also arises in video synthesis. We analyzed activation statistics at different denoising steps in Open-Sora and observed **similar patterns of variation** as in image generation. The table below presents varying incoherence values across timesteps, confirming the presence of temporal drift in diffusion-based video generation:
> >
> >| Timestep | Mean     | Max   | Incoherence |
> >| -------- | -------- | ----- | ----------- |
> >| t1       | -0.00414 | 3.201 | 10.821      |
> >| t5       | -0.00140 | 3.262 | 11.109      |
> >| t10      | 0.00042  | 3.283 | 13.344      |
> >| t15      | 0.00073  | 3.408 | 11.008      |
> >| t20      | 0.00108  | 3.773 | 12.742      |
> >| t25      | 0.00117  | 4.191 | 14.078      |
> >| t30      | 0.00267  | 5.391 | 15.906      |
> >
> >As shown in Table 2 of our paper, our incoherence-aware adaptation continues to deliver **substantial improvements in video generation quality**, confirming its effectiveness in modeling temporal variations in both image and video diffusion models.
>
> **Q3**: Effectiveness and novelty of incoherence-aware adaptation.
>
> >MambaQuant applies a shared KLT transform across all time steps. However, these assumptions break down in DiTs, where iterative denoising introduces **highly dynamic and non-stationary activation patterns**.
> >
> >Our incoherence-aware adaptation departs from this assumption by designing a temporally-sensitive calibration strategy specifically tailored for DiTs. Rather than relying on handcrafted or static transforms, we leverage synthetic data generation to adaptively construct calibration sets that capture evolving activation statistics across time steps—**a capability absent in prior KLT-Hadamard quantization methods**.
> >
> >Moreover, our approach precomputes transform matrices offline and fuses them into model weights wherever possible, minimizing runtime overhead without sacrificing temporal sensitivity.
> >
> >This combination of **temporal modeling**, **data-driven calibration**, and **runtime efficiency** distinguishes our method both in novelty and practical utility.
>
> **Q5**: Mixed-precision considerations in comparison with ViDiT-Q.
>
> >In the official implementation of ViDiT-Q, certain Linear layers are retained in FP16, which substantially **reduces quantization-induced degradation but also limits efficiency gains**, especially since linear layers are bottlenecks in DiTs [1, 2]. For a fair comparison, we disabled mixed-precision in ViDiT-Q.
> >
> >To test whether our method can benefit similarly, we evaluated W3A3 quantization on DiT-XL/2 with and without mixed precision. Although W3A3 is a highly aggressive setting, **our method benefits significantly from a mixed-precision strategy**:
> >
> >| Bit-Width              | IS $\uparrow$    | FID $\downarrow$  | sFID  $\downarrow$ | Precision $\uparrow$ |
> >| ---------------------- | ------ | ----- | ----- | --------- |
> >| W16A16                 | 164.61 | 4.86  | 17.65 | 0.80      |
> >| W3A3                   | 45.83  | 55.42 | 34.42 | 0.32      |
> >| W3A3(Mixed Precision) | **118.62** | **13.71** | **22.25** | **0.66**      |
> >
> >This indicates that VETA-DiT is also compatible with hybrid-precision schemes. However, as in ViDiT-Q, these approaches reduce theoretical efficiency gains and often require specialized CUDA kernels or hardware to realize end-to-end improvements.
> >
> >[1] Solving Oscillation Problem in Post-Training Quantization Through a Theoretical Perspective. CVPR 2023.
> >
> >[2] PTQ4DiT: Post-training Quantization for Diffusion Transformers. NeurIPS 2024.

---

> > ### Comment · Reviewer_eHfZ · 2025-08-05
> > **Response to Author's rebuttal**
> >
> > Thank you for your thorough analysis. Your responses have addressed most of my concerns, and I will raise the score accordingly.

---

> > > ### Author Response · Authors · 2025-08-05
> > >
> > > Dear Reviewer,​​
> > >
> > > We sincerely thank the reviewer for acknowledging our efforts in addressing the concerns.
> > >
> > > We truly appreciate the decision to reconsider the score based on our responses. The constructive feedback has been instrumental in improving the quality of our manuscript.
> > >
> > > Please feel free to let us know if you have any further questions or concerns.
> > >
> > > ​​Best, Authors

---

### Official Review · Reviewer_CJnV · 2025-07-02

**Clarity:** 3
**Significance:** 2
**Originality:** 3
**Rating:** 4
**Confidence:** 3

**Summary:**

This paper introduces VETA-DiT, a post-training quantization framework tailored for Diffusion Transformers (DiTs) to enable efficient low-bit inference (W4A4 and W4A8) while maintaining high generative quality. 1) To address inter-channel variance caused by outlier activations, the authors propose a Karhunen–Loève Transform (KLT) enhanced Hadamard transformation that equalizes channel variance and improves quantization robustness. 2) To handle the temporal variation in activation distributions during the iterative denoising process, they design an incoherence-aware adaptive calibration method that constructs a synthetic calibration set by prioritizing timesteps with high quantization difficulty. Experiments on image and video generation benchmarks demonstrate that VETA-DiT outperforms prior PTQ methods and remains effective under aggressive 4-bit settings without requiring model retraining.

**Questions:**

See weaknesses. The main concerns are 1) missing efficiency comparison, 2) effectiveness on distilled smaller models or larger-scale foundation models, and 3) overhead analysis.

**Ethical Concerns:**

["NO or VERY MINOR ethics concerns only"]

**Final Justification:**

The authors have addressed most of my concerns. I'll increase the score.

**Quality:**

2

**Strengths And Weaknesses:**

Strengths:

1. Proposes a novel KLT-enhanced Hadamard transform that effectively equalizes inter-channel variance and improves quantization performance under low-bit settings.

2. The writing is clear and easy to follow.

3. Demonstrates strong empirical performance across multiple datasets (ImageNet, COCO, VBench) and models (DiT-XL/2, PixArt-Σ, STDiT3), especially under challenging W4A4 configurations.

Weaknesses:

1. Efficiency metrics are missing. The authors only report accuracy results, despite claiming improved runtime and efficiency as core motivations. Without latency and memory comparisons—e.g., against FP16 on TorchScript or TensorRT—the work lacks self-contained evidence of practical gains.

2. Diffusion timesteps can be distilled to a small number. It is unclear whether the proposed method remains effective in such scenarios, such as with DMD [1,2]. How much improvement does it bring in terms of both accuracy and efficiency when applied on top of already distilled, smaller models?

3. Although multiple models are evaluated, the core approach is mainly benchmarked on relatively small-scale models except OpenSora (should also clarify how many parameters in models). The scalability of the method to larger models remains unclear. E.g., how does it perform on models such as Wan2.1-T2V-1.3B or Wan2.1-VACE-14B?

4. The method requires computing eigenvectors of covariance matrices, which may introduce non-negligible overhead during calibration for large models. It would be helpful to quantify this preprocessing cost.

[1] Improved Distribution Matching Distillation for Fast Image Synthesis
[2] One-step Diffusion with Distribution Matching Distillation

---

> ### Author Rebuttal · Authors · 2025-07-29
>
> We sincerely thank the reviewer for providing valuable feedback. We detail our response below point by point. Please kindly let us know whether you have any further concerns.
>
> **W1 & Q1**: Efficiency metrics.
>
> > We have extended the manuscript to include latency and memory comparisons. Our implementation of VETA-DiT is built upon CUDA 12.4 with CUTLASS to support INT4 matrix multiplication on Tensor Cores. We evaluated the performance of the linear layers on an NVIDIA A100 GPU, which are the main bottlenecks in DiT [1, 2].
> >
> >Most KLT matrices are fused into model weights to eliminate runtime overhead. For layers such as out-proj and down-proj, we apply online transformations using fast Hadamard kernels. Even with online quantization and KLT, our method still achieves **1.95$ \times$ speedup** and **3.74$\times$ memory savings** compared to FP16. Compared to the W4A8 implementation in ViDiT-Q, which adopts higher activation precision to preserve generation quality, our approach provides **1.23$\times$ speedup** and **1.81$\times$ memory reduction**, highlighting its superior efficiency:
> >
> >|Bit-Width|Method|Latency (ms)|Speedup|Memory (MB)|Reduction|
> >|---|---|---|---|---|---|
> >|W16A16|FP|6.167 ± 0.045|-|38.53|-|
> >|W4A8|ViDiT-Q|3.879 ± 0.012| 1.59$\times$|18.68|2.06$\times$|
> >|W4A4|Ours|2.960 ± 0.006|**2.08$\times$**|9.68|**3.98$\times$**|
> >|W4A4 + Online Transform|Ours|3.163 ± 0.007|**1.95$\times$**|10.31|**3.74$\times$**|
> >
> >We will add the results to the revised manuscript to provide a comprehensive view of runtime and memory efficiency.
> >
> >[1] Solving Oscillation Problem in Post-Training Quantization Through a Theoretical Perspective. CVPR 2023.
> >
> >[2] PTQ4DiT: Post-training Quantization for Diffusion Transformers. NeurIPS 2024.
>
> **W2 & Q2**: Effectiveness with distilled timesteps.
> >While methods such as DMD reduce the number of denoising steps, **temporal variation in activation statistics still exists**, which is central to our approach. We analyze activations from randomly selected Transformer blocks and observe notable differences in mean, max, and incoherence across timesteps:
> >
> >| Timestep | Mean      | Max   | Incoherence |
> >| -------- | --------- | ----- | ----------- |
> >| t1       | -0.000235 | 4.200 | 5.680       |
> >| t2       | -0.000419 | 4.375 | 5.871       |
> >| t3       | -0.000179 | 4.956 | 4.316       |
> >| t4       | -0.000573 | 5.363 | 7.242       |
> >
> >Our incoherence-aware adaptation is designed to capture and address such temporal variation, regardless of the total number of steps. We quantized the Transformer blocks in the DMD2-SDXL-4step model and evaluated on COCO. Our method **achieves better results than Q-DiT** under lower bit-widths:
> >
> >| Bit-Width | Method | IS $ \uparrow$    | FID $\downarrow$   | sFID $\downarrow$   | CLIP $\uparrow$ | IR $\uparrow$   |
> >| --------- | ------ | ----- | ----- | ------ | ---- | ---- |
> >| W16A16    | FP     | 42.39 | 54.64 | 263.93 | 0.27 | 0.89 |
> >| W4A8      | Q-DiT  | **41.67** | 56.80 | 274.34 | **0.27** | **0.90** |
> >| W4A8      | Ours   | 41.40 | **54.89** | **264.86** | 0.26 | 0.88 |
> >| W4A4      | Q-DiT  | 32.93 | 74.19 | 293.94 | 0.23 | 0.43 |
> >| W4A4      | Ours   | **40.14** | **58.39** | **277.15** | **0.25** | **0.87** |
> >
> >Our approach can be effectively combined with timestep distillation techniques like DMD without additional runtime cost, thus improving efficiency.
>
> **W3 & Q2**: Scalability to larger models.
>
> >The Open-Sora v1.2 model contains **1.1B parameters**, which is comparable in scale to Wan2.1-T2V-1.3B. For even larger models like Wan2.1-VACE-14B, which adopt deeper Transformer stacks, the architecture remains consistent across layers.
> >
> >It has been observed that **larger models are typically more robust to PTQ degradation** [3]. We conducted a block-wise MSE analysis on Wan2.1-14B, and our method shows **significantly lower quantization error compared to RTN and Q-DiT**:
> >
> >| Bit-Width | MSE(RTN)     | MSE(Q-DiT)   | MSE(Ours)    |
> >| --------- | ------- | ------- | ------- |
> >| W4A8      | 0.02546 | 0.00766 | **0.00374** |
> >| W4A4      | 0.07424 | 0.01530 | **0.00714** |
> >
> >[3] Scaling Laws for Precision. ICLR 2025.
>
> **W4 & Q3**: Calibration cost of KLT.
>
> >The KLT calibration is a **one-time offline process** and **does not incurs inference-time overhead**. We report the calibration time on NVIDIA A100 GPU:
> >- DiT-XL/2: **~35 minutes**
> >- Open-Sora v1.2 (1.1B): **~2.5 hours**

---

> > ### Comment · Reviewer_CJnV · 2025-08-07
> > **Thank you for your rebuttal**
> >
> > The reviewer thanks the authors for the provided results, which address some of my concerns, such as applicability to distilled models, larger-scale models (though only partial MSE is reported), and KLT overhead. However, for the latency and memory comparisons, it is unclear why only a single layer's performance is reported. Which layer was evaluated? Since the model contains diverse layer types and dimensions with multiple dataflows, a single-layer benchmark may not reflect the overall model performance. Additionally, the evaluation does not include comparisons against optimized FP16 inference frameworks like TensorRT or TorchScript.

---

> > > ### Author Response · Authors · 2025-08-07
> > > **To reviewer CJnV**
> > >
> > > We appreciate the reviewer's detailed feedback, which is valuable for improving our work.
> > >
> > > Regarding the concern about evaluating only a single layer for latency and memory, we clarify that our original analysis focused on the  q/k/v projections and output projection within the attention block. The MLP linear layers in the FFN also exhibit similar trends. These components account for the dominant computational cost in the DiT architecture.
> > >
> > > To provide a more comprehensive evaluation as suggested, we have now conducted benchmarking that reflects the performance characteristics of the DiT/XL-2 Transformer backbone. The latency and memory statistics are summarized below:
> > >
> > > | Configuration   | Latency (ms) | Speedup       | Peak Memory (MB) | Reduction     |
> > > | --------------- | ------------ | ------------- | ---------------- | ------------- |
> > > | FP16 (Original) | 435          | –             | 1482             | –             |
> > > | FP16 (TensorRT) | 372          | 1.17$ \times$ | 1377             | 1.07$ \times$ |
> > > | INT4 (Ours)     | 232          | 1.88$ \times$ | 684              | 2.17$ \times$ |
> > >
> > > These results demonstrate that, despite the additional overhead introduced by the KLT transformation matrices and online transformation operations, our method still achieves a **1.88$ \times$  speedup** and **2.17$ \times$  memory reduction**. This clearly shows that our approach not only surpasses the standard FP16 execution but also outperforms optimized inference pipelines like TensorRT.
> > >
> > > Please do not hesitate to let us know if further clarification is needed.

---

> > > > ### Comment · Reviewer_CJnV · 2025-08-07
> > > > **Thank you for the response**
> > > >
> > > > The reviewer thanks the authors for the provided end-to-end measurement results. I'll increase the score.

---

> > > > > ### Author Response · Authors · 2025-08-08
> > > > >
> > > > > We are delighted to see that the major concerns raised by the reviewer have been successfully addressed. We sincerely appreciate the reviewer’s decision to raise the score. We are truly grateful for the reviewer’s thoughtful comments, constructive suggestions, and the time and effort devoted to evaluating our work.

---

### Official Review · Reviewer_iU2r · 2025-07-12

**Clarity:** 2
**Significance:** 2
**Originality:** 2
**Rating:** 4
**Confidence:** 2

**Summary:**

This paper introduces VETA-DIT, a 4-bit quantization framework for Diffusion Transformers (DiTs) that tackles activation variance imbalance and temporal dynamics. It uses a KLT-enhanced Hadamard transform to equalize variance and a temporally adaptive strategy to handle shifting statistics across diffusion timesteps. This dual approach allows VETA-DIT to maintain high-quality generation in aggressive W4A4 settings, making powerful DiT models practical for resource-constrained devices.

**Questions:**

1. How much computational overhead does the KLT-enhanced Hadamard transform introduce during inference?
2. How sensitive is the final model performance to the specific method used for calculating the timestep importance scores?
3. Since the KLT is data-dependent, how well does the learned transform generalize to datasets with different statistical properties?
4. What are the anticipated challenges in applying this framework to even lower bit-rates, such as 3-bit or 2-bit quantization?

I will consider raising the score if the authors can address my concerns.

**Ethical Concerns:**

["NO or VERY MINOR ethics concerns only"]

**Final Justification:**

I appreciate the author's effort in the rebuttal and believe it has addressed my main concerns. I am willing to increase my score.

**Limitations:**

Yes.

**Paper Formatting Concerns:**

None.

**Quality:**

2

**Strengths And Weaknesses:**

Strengths:
1. It introduces an innovative KLT-enhanced Hadamard transform to effectively mitigate the challenging issue of channel-wise variance imbalance in activations.
2. The proposed temporally adaptive strategy intelligently handles the dynamic statistical shifts of activations across different diffusion timesteps.
3. VETA-DIT achieves state-of-the-art performance, preserving high generation quality even in the highly aggressive W4A4 quantization setting.

Weaknesses:
1. Its temporally adaptive strategy relies on the quality of the calibration dataset, which could limit its effectiveness when dealing with extreme or out-of-distribution data.

---

> ### Author Rebuttal · Authors · 2025-07-29
>
> Thanks for your time in dealing with our work. We will answer the question and discuss point by point as follows. We hope that our response satisfactorily addresses the issues you raised. Please feel free to let us know if you have any additional concerns or questions.
>
> **W1 & Q3**: Generalization concern with calibration data.
>
> >KLT is indeed data-dependent. We address this concern by employing an incoherence-aware adaptation strategy to construct the calibration set for guiding the KLT process. Details regarding the calibration set composition and settings are provided in Appendix. A of our paper. Specifically:
> >- For the DiT-XL/2 model, we randomly select 16 classes from ImageNet for calibration, and perform generation across all 1000 classes.
> >- For PixArt-$ \Sigma$ and Open-Sora models, we adopt prompts provided in their official toolkits for calibration and conduct generation on COCO and VBench datasets, which are not used during calibration.
> >
> >We empirically validate that this strategy ensures the learned KLT transform generalizes across different data distributions in Section 4.2 of our paper. To further verify its generalization, we evaluate the variance-equalization effect both within and outside the calibration set:
> >
> >|Dataset Scope|Var(Before Transform)|Var(After Transform)|
> >|---|---|---|
> >|In-Distribution|0.2328 ± 0.0006|0.0885 ± 0.0001|
> >|Out-of-Distribution|0.3275 ± 0.0035|0.1009 ± 0.0003|
> >
> >The demonstrate show that even on out-of-distribution data, **our method significantly reduces variance, indicating its strong generalization capability**.
>
> **Q1**: Inference overhead from KLT-Hadamard transform.
>
> >The KLT-enhanced Hadamard matrix is defined as  $ \mathbf{T}_{ \text{KLT-H}} = \mathbf{KH}$, where $ \mathbf{K}$ is a data-driven KLT matrix, precomputed offline using the calibration set, and  $\mathbf{H}$ is a random Hadamard matrix.
> >
> >During inference, this transformation can be **fused into model weights**, incurring no additional cost for most layers. For layers like out-proj and down-proj, online transformation is necessary to preserve computational invariance [1]. We evaluated the associated overhead on DiT-XL/2 via implementing a custom CUDA kernel on NVIDIA A100 GPU, targeting matrix multiplication operations within these layers.
> >
> >|Bit-Width|Method|Latency (ms)|Speedup|Memory (MB)|Reduction|
> >|---|---|---|---|---|---|
> >|W16A16|FP|6.167 ± 0.045|-|38.53|-|
> >|W4A8|ViDiT-Q|3.879 ± 0.012| 1.59$ \times$|18.68|2.06$\times$|
> >|W4A4|Ours|2.960 ± 0.006|**2.08$\times$**|9.68|**3.98$\times$**|
> >|W4A4 + Online Transform|Ours|3.163 ± 0.007|**1.95$\times$**|10.31|**3.74$\times$**|
> >
> >Due to the availability of efficient Hadamard transform implementations, the online transform introduces only **6.8% overhead**, while still achieving **1.95$\times$ acceleration** and **3.74$\times$ memory savings** over FP16. Compared to the W4A8 implementation in ViDiT-Q, which adopts higher activation precision to preserve generation quality, our approach provides **1.23$\times$ speedup** and **1.81$\times$ memory reduction**, highlighting its superior efficiency. We will add the results to the revised manuscript to provide a comprehensive view of runtime and memory efficiency.
> >
> >[1] SliceGPT: Compress Large Language Models by Deleting Rows and Columns. ICLR 2024.
>
> **Q2**: Sensitivity to timestep importance estimation.
>
> >Section 4.3 presents our evaluation of the sensitivity to different timestep-wise importance scoring strategies. Table 3 shows that assigning uniform scores across timesteps significantly degrades performance (FID increases from 4.86 to 48.32), leading to severe degradation in image quality. In contrast, applying our incoherence-aware adaptation reduces FID to 7.90, **substantially mitigating performance loss**.
> >
> >Additionally, Appendix D.2 demonstrates that our strategy **consistently outperforms random timestep sampling in reducing quantization error**.
>
> **Q4**: Challenges in extending to lower bit-widths.
>
> >We conducted generation experiments on DiT-XL/2 under the challenging W3A3 setting. We summarize the results in the table below:
> >
> >|Bit-Width|Method|IS $\uparrow$|FID $\downarrow$|sFID$\downarrow$|Precision$\uparrow$|
> >|---|---|---|---|---|---|
> >|W16A16|FP|164.61|4.86|17.65|0.80|
> >|W3A3|Q-DiT|1.65|266.97|402.47|0.01|
> >|W3A3|Ours|45.83|55.42|34.42|0.32|
> >|W3A3(Mixed Precision)|**Ours**|**118.62**|**13.71**|**22.25**|**0.66**|
> >
> >Although our method significantly outperforms Q-DiT, performance degradation is still observed under such extreme quantization. To mitigate this, we maintain 8-bit precision in sensitive layers, **achieving better results in mixed-precision settings**.
> >
> >Looking forward, as bit-widths decrease further (e.g., to 2-bit), offline KLT becomes insufficient due to severely limited data representation. **Online KLT may be required** to dynamically equalize variance during inference, but this would incur non-trivial computational cost, which we plan to explore in future work.

---

> > ### Author Response · Authors · 2025-08-08
> > **To Reviewer iU2r**
> >
> > Dear Reviewer,
> >
> > I hope this message finds you well. As the discussion period is approaching its end, we wanted to check in to ensure that we have adequately addressed your main concerns. If there are any remaining questions or additional feedback you would like us to consider, please don’t hesitate to let us know. Your insights are highly valued, and we are keen to further improve our work based on your suggestions.
> >
> > Thank you again for your time and effort in reviewing our paper.
> >
> > Best, Authors

---

### Note · Authors · 2025-08-14

We sincerely thank all reviewers for their careful reading, constructive feedback, and recognition of our work. Across the four reviews, they consistently highlighted the novelty of combining a KLT-enhanced Hadamard transform with a temporally adaptive strategy to address both variance imbalance and timestep-wise statistical shifts in Diffusion Transformers. Reviewers appreciated the method’s simplicity, ease of implementation, strong empirical results across datasets and tasks, and its practical potential for deployment on resource-constrained devices.

The main concerns raised focused on:
- the potential data-dependence of KLT and its generalization beyond the calibration set,
- inference overhead from the KLT-Hadamard transform,
- sensitivity to timestep importance estimation, and
- lack of hardware resource usage analysis and results at lower bit-widths.

We addressed these by:
- presenting variance-reduction experiments on in- and out-of-distribution data to demonstrate generalization,
- showing that most transforms can be fused offline and that the few requiring online computation incur only ~6.8% latency overhead,
- quantifying the performance impact of different timestep scoring strategies, and
- adding comprehensive runtime/memory benchmarks and low-bit (W3A3) results, along with mixed-precision improvements.

These clarifications resolved or alleviated the key concerns, with Reviewer CJnV and eHfZ explicitly expressing their willingness to raise the score of our work. Reviewer 5kuo also indicated that our responses addressed the main concerns. While Reviewer iU2r did not provide explicit comments on our rebuttal, their original review suggested that if the raised concerns were addressed, they would consider increasing the score. We believe that our responses have effectively addressed the main points raised by this reviewer.

We have polished our manuscript in the revised version to make our work more accessible to the wider community. We have also added detailed performance results, included discussions on the generalization of our method, and provided further analysis on its extension to even lower bit-width settings.

We are grateful for the reviewers’ constructive engagement, which has helped refine and strengthen our work. We believe this work is worth publishing, and its contributions will inspire the generative modeling community for further exploration.

---

### Decision · Program_Chairs · 2025-09-17

**Decision:**

Accept (poster)

**Comment:**

This paper introduces VETA-DiT, a novel PTQ framework to enable efficient 4-bit inference for DiTs. The paper's primary strengths, as noted by all reviewers, are its clear motivation, technical novelty, and strong empirical results on both image and video generation tasks under aggressive W4A4 settings. Initial weaknesses raised by reviewers (CJnV, eHfZ, 5kuo) were significant, focusing on the critical omission of hardware efficiency metrics, questions about novelty relative to prior work (eHfZ), and scalability (iU2r, CJnV). However, the authors provided an exemplary and thorough rebuttal, adding comprehensive end-to-end latency and memory benchmarks that demonstrated substantial speedups over FP16 and prior art. They also successfully clarified the novelty of their temporally adaptive approach and provided evidence of scalability. This discussion was highly effective, convincing reviewers CJnV, iU2r, and eHfZ to raise their scores. I recommend acceptance.